



# Towards a Global Spatial Machine Learning Model for Seasonal Groundwater Level Predictions in Germany

Stefan Kunz[1], Alexander Schulz[2], Maria Wetzel[1], Maximilian Nölscher[1], Teodor Chiaburu[2], Felix Biessmann[2], and Stefan Broda[1]

[1]Federal Institute for Geosciences and Natural Resources, Berlin, Germany
[2]Berliner Hochschule für Technik , Berlin, Germany

**Correspondence:** Stefan Kunz (Stefan.Kunz@bgr.de)

**Abstract.** Reliable predictions of groundwater levels are crucial for a sustainable groundwater resource management, which needs to balance diverse water needs and to address potential ecological consequences of groundwater depletion. Machine Learning (ML) approaches for time series prediction, in particular, have shown promising predictive accuracy for groundwater level prediction and have scalability advantages over traditional numerical methods when sufficient data is available. Global

ML architectures enable predictions across numerous monitoring wells concurrently using a single model, allowing predictions for monitoring wells over a broad range of hydrogeological and meteorological conditions and simplifying model management. In this contribution, groundwater levels were predicted up to 12 weeks for 5,288 monitoring wells across Germany using two state-of-the-art ML approaches, the Temporal Fusion Transformer (TFT) and the Neural Hierarchical Forecasting for Time Series (N-HiTS) algorithm. The models were provided with historical groundwater levels, meteorological features and a wide

range of static features describing hydrogeological and soil properties at the wells. To determine the conditions under which the model achieves good performance and whether it aligns with hydrogeological system understanding, the model's performance was evaluated spatially and correlations with both static input features and time-series features from hydrograph data were examined.

  The N-HiTS model outperformed the TFT model, achieving a median NSE of 0.5 for the 12-week prediction over all 5,288

monitoring wells. Performance varied widely: 25% of wells achieved an NSE > 0.68, while 15% had an NSE < 0 with the best N-HiTS model. A tendency for better predictions in areas with high data density was observed. Moreover, the models achieved higher performance in lowland areas with distinct seasonal groundwater dynamics, in monitoring wells located in porous aquifers, and at sites with moderate permeabilities, which aligns with theoretical expectations. Overall, the findings highlight that global ML models can facilitate accurate seasonal groundwater predictions over large, hydrogeological diverse

areas, potentially informing future groundwater management practices at a national scale.



## 1 Introduction

The growing availability of large datasets for hydrogeological applications has led to an increased use of machine learning (ML) approaches in the hydrogeological domain for tasks such as groundwater level prediction (Tao et al., 2022), as well as related tasks such as predicting groundwater recharge rates (Jung et al., 2024) or the 3D characterisation of aquifer systems (Manzoni et al., 2023). In recent years, a growing body of studies demonstrated the potential of ML methods for groundwater level prediction, both in smaller areas and for monitoring wells distributed throughout Germany (Heudorfer et al., 2024; Wunsch et al., 2022a, 2021, 2018; Guzman et al., 2017). ML approaches aim to implicitly learn the underlying dynamics from measured data by modelling the statistical relationships between input features (e.g. precipitation) and target features (e.g. groundwater levels). These models are based on the data at the respective groundwater monitoring wells, and thus easy to scale spatially and effective for groundwater level prediction over larger areas.

Single-well (local) models are the state-of-the-art in ML-based groundwater level prediction, where for each monitoring well, an individual model has to be trained. In contrast, the class of so-called "global ML models" allows the creation of one ML model capable of predicting groundwater levels for multiple monitoring wells. These models train on different time series to learn underlying patterns shared across multiple time series potentially enhancing the overall forecasting performance. Compared to single-well models, using multiple monitoring wells at once increases the diversity of training data, thereby widening the training envelope which can result in improved performance during inference, e.g. because the model had access to rare observations such as extreme precipitation events (Kratzert et al., 2024). Furthermore, global ML approaches simplify model development and maintenance in comparison to single-well models. Besides incorporating time-dependent (dynamic) input features such as precipitation and temperature, these ML approaches can also make use of temporally static features that describe physical or environmental properties at the monitoring sites (e.g., permeability coefficient). These static features enable information sharing within groups of time series with similar static variable levels, aiming to enhance the model's ability to generalise to locations with similar static feature levels. By incorporating static features, these global models possess the potential for generalisation and regionalisation (Heudorfer et al. (2024); Kratzert et al. (2019)). To assess the uncertainty in the predicted target feature from ML approaches for time series prediction different methods are used, such as ensembling, Monte Carlo Dropout (Althoff et al., 2021), quantile forecasting (Kan et al., 2022), or directly predicting a distribution of the target feature (Klotz et al., 2022), while in numerical groundwater models often stochastic model calibration methods are employed, which may involve sensitivity analyses of model parameters (Manzoni et al., 2024; Linde et al., 2017).

So far, in forecasting competitions, such as M4 (Makridakis et al., 2020) and M5 (Makridakis et al., 2022), global time series models were successfully applied to forecasting problems in finance, retail, and economics and outperformed their competing local models. Recently, several new neural network architectures for global time series prediction were proposed. Among these architectures are DSSM (Rangapuram et al., 2018), DeepGLO (Sen et al., 2019), DeepAR (Salinas et al., 2020), StemGNN (Cao et al., 2020), TFT (Lim et al., 2021), N-HiTS (Challu et al., 2022), and TiDE (Das et al., 2024). The recently developed TFT, a combination of recurrent neural networks and self-attention layers, and N-HiTS, a time-series decomposition algorithm based on Multilayer Perceptrons (MLPs), have shown promising predictive capabilities. The TFT architecture has been already



successfully applied in the environmental sciences, e.g., for forecasting $CO_2$ Emissions from one hour up to one week using about 6 Million $CO_2$ measurements and other time-dependent (dynamic) variables (Linardatos et al., 2023). Furthermore, the TFT was also used for streamflow prediction on 2610 basins across the world using 38 dynamic and 211 static features, thereby outperforming models based on Long-Short Term Memory (LSTM) and other Transformer architectures (Koya and Roy, 2023). In environmental research, the N-HiTS architecture has been successfully applied to predict wastewater levels in

rain tanks over a ten hours prediction horizon (Chiaburu and Bießmann, 2024). Furthermore, N-HiTS was compared to other models in a study on global weather forecasting (Wu et al., 2023).

For predicting groundwater levels in Germany, a recent study by Heudorfer et al. (2024) used a global LSTM architecture integrating both dynamic (meteorological) and static input features. Their optimal models achieved a Nash-Sutcliffe Efficiency (NSE) of above 0.8 in the test period across a selected set of 108 monitoring wells, which were relatively evenly distributed

throughout Germany. This study seeks to expand upon this research by using a larger number of monitoring wells and exploring more complex ML architectures. In this study, the two state-of-the-art global ML time series architectures TFT and N-HiTS are applied to seasonally predict groundwater levels up to 12 weeks for the so far most comprehensive groundwater data set covering large parts of Germany (5,288 monitoring wells). To the best of our knowledge, this is the first study that attempts to make groundwater level predictions on a national scale for such a high number of monitoring wells and one of the first studies

that uses global ML architectures to predict groundwater levels. We hypothesise that the large amount of time series data used in combination with static input features will help to achieve a good predictive performance and enhance hydrogeological system understanding.

The remainder of the paper is organised as follows: first we introduce the dataset (Section 2.1) and the deep learning architectures (Section 2.2) along with the experimental design (Section 2.3). Then, the predictive performance is reported for both

architectures and analysed with respect to factors affecting the predictive performance. To quantify the impact of static features on the prediction accuracy, model variants with and without static features are examined for both architectures. Moreover, the interpretable nature of the TFT is used to understand the importance of the input features and important past time steps for the model performance. Finally, the results are discussed in terms of general and spatial model performance as well as the feature importance to demonstrate the potential of modern ML approaches for groundwater level prediction and to determine if the

models reflect our understanding of the hydrogeological system (Sections 3 and 4).



## 2 Data and methods

### 2.1 Data

#### 2.1.1 Groundwater level time series and preprocessing

Groundwater level measurements in the period from 1990 to 2016 were provided by the environmental agencies and the geo-
logical surveys of most of the German federal states. The preprocessing of the groundwater levels was kept simple mainly for
two reasons; first simple and few preprocessing steps allow to retain as many training data points as possible, second it allows to
assess the predictive performance under realistic conditions with heterogeneous data distributions. For each groundwater level
time series jumps between two time points greater than 50 times the average change of the given time series were removed
and gaps up to four weeks were linearly interpolated. Time series with gaps greater than four weeks were truncated at the gap.
For every well, groundwater level time series were split into training, validation and test data set. Validation and test data set
comprise a period of 3 years starting in 2010 and 2013, respectively. The training data only included time series that were at
least six years long and available from 2004 onwards to the validation period starting in 2010 (Figure 1 A). Thus, the length of
the training period varied (descriptive statistics in Supplement Table A1), whereby 54 % of all time series spanned the entire
period of available data until 1990 (Figure 1B).

The preprocessing resulted in groundwater time series of 5,288 monitoring wells, which corresponds to approximately 4.5
million records for model training. These monitoring wells are distributed throughout Germany, with the exception of some
federal states, where only few (Mecklenburg-Western Pomerania, Bremen, Hamburg) or no monitoring wells (Thuringia and
Saarland) satisfied the preprocessing criteria.

#### 2.1.2 Dynamic features

An overview of the dynamic input features can be found in Table 1. Besides being the target to predict, the groundwater level
was also used as a dynamic input feature. Additionally, meteorological forcings which have a strong influence on groundwater
levels were used. Hence, as further dynamic input features, the meteorological variables precipitation (mm), relative humidity
(%), and temperature (°C) were selected and extracted from HYRAS 5.0 (Razafimaharo et al., 2020). Moreover, a sinusoidal
curve fitted to the temperature was used, which was a good predictor for groundwater levels in previous studies (Wunsch
et al., 2021). The HYRAS data set is a raster product published by the National German Weather Service and holds gridded
meteorological data for Germany over the period from 1951 to 2022 for temperature, to 2023 for humidity, and to 2024 for
precipitation (status as of October 2024). Additional dynamic input features were the leaf area index (LAI) (Pistocchi, 2015)
and the day of the year expressed in sine and cosine values.

The meteorological input features and the LAI were extracted within a 1 km buffer around the groundwater wells. Thereby,
a weighted average was calculated based on the area covered by the pixels within the buffer. Dynamic features were divided
into training, validation and test data based on the split of the groundwater level time series.



### 2.1.3 Static features

The static features used in the study are environmental characteristics from the domains hydrogeology, soil, topography and land cover (see Table 2). They describe dominant physical factors that control groundwater dynamics at the well locations

aiming to facilitate the generalisation of relationships between monitoring sites with similar environmental characteristics. All static input features except the Landform Shannon Index and the elevation were extracted within a 1 km buffer around the groundwater wells. For the values of the numerical features, a weighted average was calculated based on the area covered by the pixels within the buffer. For categorical features, the category with the largest area share within the buffer was used. Land-form Shannon Index and elevation were extracted directly at the well. The Topographic Wetness Index (TWI) was calculated

according to Beven and Kirkby (1979). Additionally, the mean and the standard deviation of each groundwater level time series computed on the respective training data periods were used as static input time-series features.

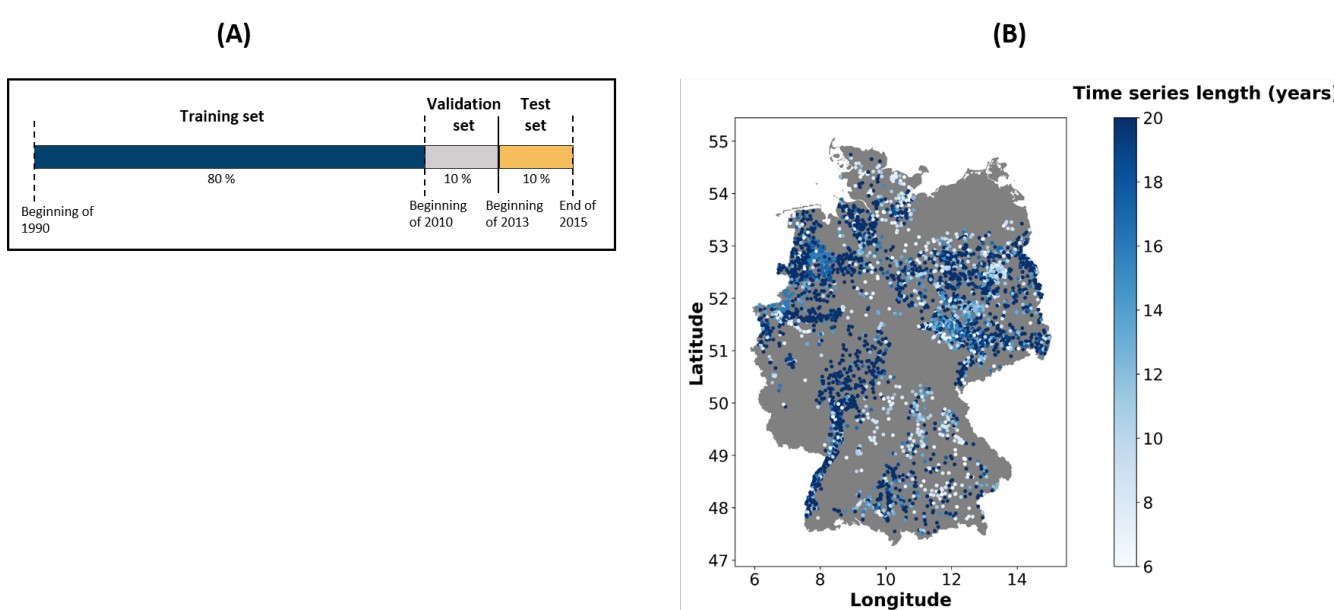

**Figure 1.** (A) Split in training, -validation and -test data. (B) Spatial distribution of the 5,288 wells used for groundwater level prediction. The colour scale indicates the length of the individual groundwater level time series in the training dataset.



**Table 1.** Description of the dynamic input features.

| Type | Name | Description | Source |
|---|---|---|---|
| Hydrogeology | Groundwater levels | Groundwater hydrographs aggregated to weekly values. | Provided by the environmental agencies and geological surveys in Germany (unpublished) |
| Meteorology | Temperature | Mean temperature (daily values aggregated to weekly) | Razafimaharo et al. (2020) |
| | Sinus Temperature | - | |
| | Precipitation | Sum precipitation (daily values aggregated to weekly) | Razafimaharo et al. (2020) |
| | Relative humidity | Mean relative humidity (daily values aggregated to weekly) | Razafimaharo et al. (2020) |
| Vegetation | Leaf Area Index (LAI) | The LAI describes the area of leaves per unit ground area and is related to the evapotranspiration in an area. It is reported in monthly averaged values. | Pistocchi (2015) |
| Annual cycle | Sinus Day | - | |
| | Cosinus Day | | |



**Table 2.** Description of the static input features.

| Type | Name | Description | Source |
|---|---|---|---|
| Hydrogeology | Hydrogeological Spatial Structure of Germany | Classification of areas with similar hydrogeological characteristics, groundwater conditions and geologic genesis in Germany. The ten major hydrogeological districts are used. | BGR and SGD (2019) |
| | Aquifer type | Five aquifer type categories (e.g. porous or karstified) | BGR and SGD (2019) |
| | Permeability coefficient | Six categories (e.g. high (>0.001 - 0.01 m/s)) | BGR and SGD (2019) |
| | Groundwater recharge | Mean annual groundwater recharge rates 1961-1990 (mm/a) | BGR (2019) |
| | Mean groundwater level | - | |
| | Standard deviation groundwater level | - | |
| Soil | Soil texture | 13 categories (e.g. sandy loam) | BGR (2007) |
| Land cover | Land cover | Nine categories (e.g. herbaceous vegetation) | Buchhorn et al. (2017) |
| Topography | Topographic Wetness Index (TWI) | Terrain driven wetness potential | Beven and Kirkby (1979) |
| | Elevation | Elevation at the groundwater well | European Environment Agency (2018) |
| | Shannon Index Landforms | Diversity index of landform types. Higher values indicate more landforms types and/or landform types having more similar proportions within the aggregation window (10 x 10 km) | Amatulli et al. (2018) |
| | Divide to stream distance (EUMOHP_DSD1) | Distance from hypothetical groundwater catchment divide to nearest stream (hydrologic order 1) at the monitoring well location | Nölscher et al. (2022) |
| | Lateral position (EU-MOHP_LP1) | Relative position of the monitoring well lateral along the divide-to-stream stretch (hydrologic order 1) | Nölscher et al. (2022) |
| | Stream distance (EU-MOHP_DS1) | Distance from the monitoring well to the nearest stream (hydrologic order 1) | Nölscher et al. (2022) |





## 2.2 Global Machine Learning Algorithms

### 2.2.1 Temporal Fusion Transformer (TFT)

The TFT is an attention-based neural network architecture for time series forecasting and has achieved good results on various
forecasting tasks, often outperforming other models like LSTMs and vanilla Transformers (Lim et al., 2021). The main building
blocks of the TFT are gating mechanisms, Variable Selection Networks (VSN), static covariate encoders, LSTM encoder-
decoders and multi-head self-attention layers.

The gating mechanisms regulate the degree of non-linear processing of the input by employing Gated Resiual Networks
(GRN), allowing the model to skip entire layers when necessary. VSN help to remove noisy inputs and can be used to assess
the importance of input features. Thereby, feature importance is assessed separately for static and dynamic inputs (i.e., past
and future inputs) on the test set. At each time step, a flattened features vector is fed into a GRN followed by a Softmax layer
producing an output vector with so-called variable selection weights. The importance of each feature is then represented by
the average of the variable selection weights over all time steps. After the VSN is applied, static covariate encoders integrate
static information with the dynamic information using a GRN. The dynamic outputs of the VSN are passed to the LSTM
encoder-decoder which captures short-term dependencies by learning past data representations (encoder) to predict future
values (decoder). Self-attention is then applied to the output by the LSTM encoder-decoder to weigh the different time steps of
input sequences and dynamically adjust how they affect the output. Self-attention layers enable the model to identify important
time steps and approximate long-term dependencies across the input sequences. The attention weights can be visualized for
interpretability.

### 2.2.2 Neural Hierarchical Interpolation for Time Series Forecasting (N-HiTS)

N-HiTS (Challu et al., 2022) is a state-of-the-art deep learning model for time series forecasting, improving upon its predeces-
sors the N-BEATS and N-BEATSx models (Oreshkin et al., 2020; Olivares et al., 2022), by enhancing long-term forecasting
capabilites and computational efficiency. N-HiTS outperformed its predecessors and several transformer-based methods on
various forecasting tasks, especially for long-term horizons, and also required less memory and computational power (Challu
et al., 2022).

The N-HiTS architecture is composed of so-called blocks organized into different stacks. Each block consists of a Max-
Pool layer and several Multilayer Perceptrons MLPs. The MaxPool layers subsample the series at different resolutions (e.g.,
daily, weekly) allowing each stack to focus on different frequencies (short-term to long-term) within the data. Frequencies
are projected onto a basis of simple functions such as sine waves and step functions (related concepts are Short Time Fourier
Transform or Wavelets). Each subsampled version of the series is processed by a series of MLPs that learn patterns within the
data at their respective frequencies. Each MLP block outputs a backcast and forecast, where the backcast is subtracted from the
input and the remaining signal is passed to the next block through residual connections. The partial predictions obtained from
each block are combined through hierarchical interpolation. The final predictions consist of the summed-up partial predictions
from each stack.



## 2.3 Experimental Design and Training

In order to assess the impact of the static features on the predictive quality of the global models, two model variations were considered: the models were provided with 1) all dynamic and static features, and 2) solely dynamic features (referred to as purely dynamic models hereafter).

Groundwater levels were predicted from one up to 12 weeks. For every time step, a look-back window (i.e. sequence length) for the dynamic features of 52 weeks was used to represent one annual cycle. All model variants were trained for ten epochs and with ten different random seeds to account for the stochasticity in the initialisation of model weights. Large batch sizes were used (TFT: 4096, N-HiTS: 1024) to avoid overfitting and to accelerate the training. The risk of overfitting was further reduced by the application of early stopping on the validation loss, a dropout rate of 0.2, and learning rate scheduling using stochastic weight averaging after the second epoch (Izmailov et al., 2019). During training, models were optimised with the Ranger optimiser (Wright and Demeure, 2021). For both architectures the multivariate quantile loss function was used (Kan et al., 2022). The quantile loss was chosen because it is more robust towards outliers than for example the root mean squared error (RMSE), e.g., caused by extreme precipitation events.

## 2.4 Model evaluation

For the predicted groundwater levels, the 0.5 quantile is reported as point forecast. The prediction intervals are based on the 0.10 and 0.90 quantiles. All models were evaluated with the Nash-Sutcliffe efficiency (NSE) on the test data. In addition, the root mean square error (RMSE), the relative Mean Bias Error (rMBE), and, for evaluating prediction intervals, the Interval Score are reported for comparison. The rMBE is defined as the mean deviance divided by a constant, for which the standard deviation of the groundwater levels is chosen. The Interval Score is a proper scoring rule that combines two properties of an prediction interval: (1) the sharpness, which refers to the concentration of the predictive distribution and is a property of the prediction only and (2) the calibration, which denotes how much the true value is outside of the predicted interval (Gneiting and Raftery, 2007). Smaller values indicate a better estimate and a narrower prediction interval. Details on the calculation of both metrics is given in the supplement subsection A2). To compare model performances, we aggregated the error metrics to the median for each well across the 10 initialisations. Additionally, we also show the median NSE values across the 5,288 monitoring wells for each individual initialisation (Figure 2 B).

To investigate if certain hydrogeological conditions allow better predictions, the NSE values of each monitoring well were evaluated for a prediction horizon of 12 weeks across the categorical static features. Additionally, Spearman correlations of the static numeric features and the NSE of each monitoring well were calculated. Areas with a high data density were identified using Kernel Density Estimation (KDE) and median NSE values for the monitoring wells within these areas were compared.

To examine how the ML approaches performed with regard to groundwater hydrographs exhibiting differing behaviours, NSE values were correlated with a number of time series features describing the groundwater dynamics at each monitoring well. The time series features used were groundwater hydrograph specific features derived by Wunsch et al. (2022b), namely the seasonal behaviour (agreement with the expected seasonality, i.e. Max in March and Min in September), the flashiness





(frequency of short-term changes), and the range ratio (superimposed long-period signals). The calculation of these features followed Wunsch et al. (2022b). Furthermore, standard statistical time series features were employed, namely the amplitude (range), skewness, linear trend (slope) and length of the time series (including the test period). For the linear trend, a linear regression was fitted and the obtained slope was correlated with the NSE, whereby non-significant slopes were set to zero. The non-parametric Spearman correlation coefficient was used, given that the NSE and the features considered for correlation analysis did not seem to satisfy the bivariate normality assumption of the Pearson correlation coefficient.

The TFT architecture offers interpretable insights via the variable selection weights and attention scores. For the predictive performance of the TFT, the feature importance ranking is reported as well as important past time steps.

## 3 Results

### 3.1 General model performance

Both global ML architectures achieved a high performance for the one-week prediction horizon with a median NSE > 0.9 for all 5,288 monitoring wells. For the 12-week prediction horizon, the N-HiTS models consistently outperformed the TFT models based on the NSE. The N-HiTS model provided with static and dynamic features achieved a median NSE of 0.5 across the 5,288 monitoring wells (TFT 0.34, Figure 2A, Supplement Table B1), meaning that for approximately 81 % of the monitoring wells (4286 monitoring wells) it achieved higher NSE values than the TFT. A wide spread in the model performance was observed for all model variants, e.g., the N-HiTS model with static and dynamic input features achieved for 25 % of the monitoring wells an NSE of 0.68 or greater (TFT 0.55 or greater). On the other hand, an NSE below zero was achieved for about 15 % of the monitoring wells, indicating that the model predictions were a worse approximation to the ground truth than the mean of the observed values (TFT 22 % of the monitoring wells). Despite relatively similar NSE values at the one-week prediction horizon, the NSE decreased more prominently for the TFT model variants than for the N-HiTS model variants over the longer prediction horizons. The performance of the TFT models was more influenced by the initialisation of the model weights compared to the N-HiTS models, indicated by the larger spread of median NSE values per initialisation (Figure 3).

In terms of the RMSE and rMBE the N-HiTS models performed slightly better than the TFT models (Figure 2A, Figure 2B). For the 12-week prediction horizon, N-HiTS provided with static and dynamic features achieved a median rMBE of 0.13 meters (TFT 0.15 meters) and median rMBE of 0.08 meters (TFT 0.13 meters). For the RMSE and rMBE, the differences between the purely dynamic model variants and the models provided with all dynamic and static input features were mostly small around 0.01 meters.

Regarding the Interval Score (Figure 2B), the N-HiTS models achieved lower values than the TFT models, indicating that smaller and more precise prediction intervals could be obtained with N-HiTS. For the 12-week prediction horizon, the purely dynamic model variants had smaller Interval Scores than the model variants provided with static and dynamic input features (Table B2). N-HiTS (purely dynamic) achieved an Interval Score of 3.68 (TFT (purely dynamic) 4.41), while the Interval Score for the N-HiTS model provided with static and dynamic features was 3.89.




The addition of static features marginally improved the performance of the N-HiTS architecture (difference in median NSE = 0.03 to the purely dynamic model for the 12-week prediction, P < 0.001) and did not improve the performance of the TFT architecture (Figure 2A, Supplement Table B1).

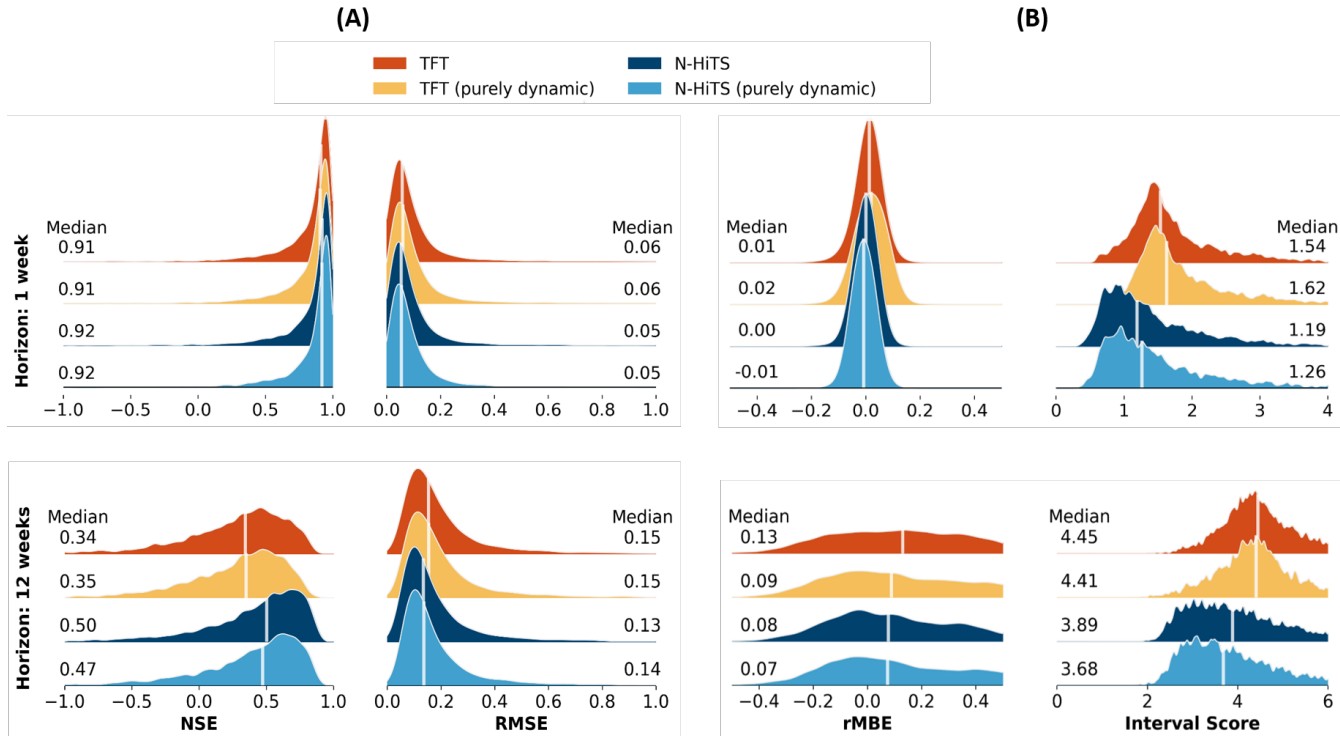

**Figure 2.** Overview on the predictive performance of the global models. A) Distributions of NSE and RMSE values achieved for the 5,288 monitoring wells. Reported values are based on the median NSE and RMSE of the 10 initialisations for each well. White lines denote the median value of each distribution. B) Similar depiction for the achieved rMBE and Interval Score values of the 5,288 wells for the model variants with all features and the purely dynamic model variants.



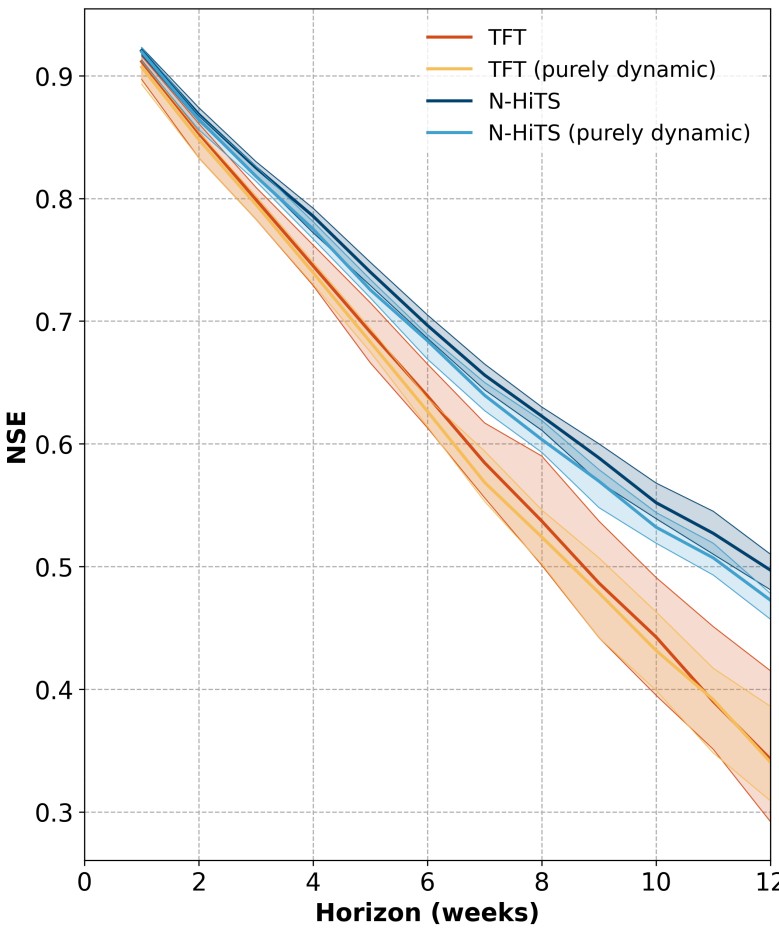

**Figure 3.** Range of Median NSE values of the ten initialisations (shaded colors) for each model variant for the 12 prediction horizons. The bold lines represent the median of the ten initialisations.





## 3.2 Spatial model performance

Investigating how the prediction accuracy is distributed among the groundwater monitoring wells reveals spatial patterns and
variations across Germany. Both, TFT and N-HiTS models showed similar spatial patterns in the predictive performance (Supplement Figure B2). For the sake of simplicity, we focus on the results of the best-performing model, the N-HiTS model provided with static and dynamic features. As mentioned earlier, for the one-week prediction horizon there were almost exclusively forecasts with a high prediction accuracy. Poor performances with a median NSE below zero were observed for wells primarily located in South-East Germany. For the 12-week horizon, there are regions where the model forecasts appear to be
generally better and above the median NSE of 0.5, particularly in the Upper Rhine Graben, North-West Germany and large parts of North-East Germany (Figure 4, Supplement Figure B2). Notably, better predictions were often achieved in areas with higher data density, though exceptions exist. The KDE analysis identified eight areas with high data density (Figure 5). Among them, five show a better performance than the median NSE of 0.5 (Median NSE between 0.59 and 0.78). However, three high data density areas, located in the southeast of the North and Central German Unconsolidated Rock District, central south of
the North and Central German Unconsolidated Rock District, and in the west of the Alpine Foreland, had median NSE values between 0.29 and 0.32. The Spearman correlation between the NSE and the KDE values was 0.12. Regarding the model performance of other static feature categories, e.g. soil texture categories, there was no evidence that the model performance was clearly better under certain conditions. Overall, there was no clear visual spatial distribution of NSE values, i.e. monitoring wells with poor model prediction accuracy (NSE below 0) were distributed across Germany and also occurred in close
proximity to sites with better prediction accuracy, resulting in an uneven spatial pattern of model performance.





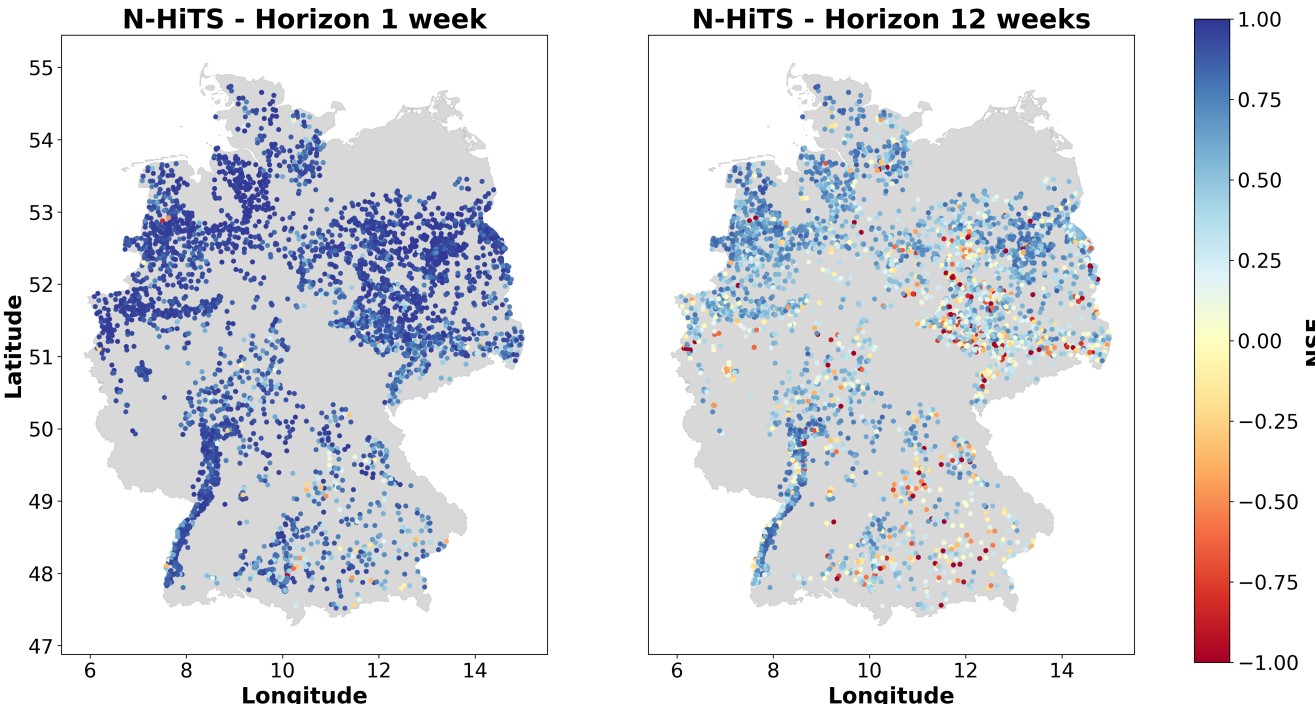

**Figure 4.** Performance of the N-HiTS model in terms of NSE across Germany for the one-week prediction and 12-week prediction horizon. Results for the TFT model are shown in the Supplement (Figure B2).





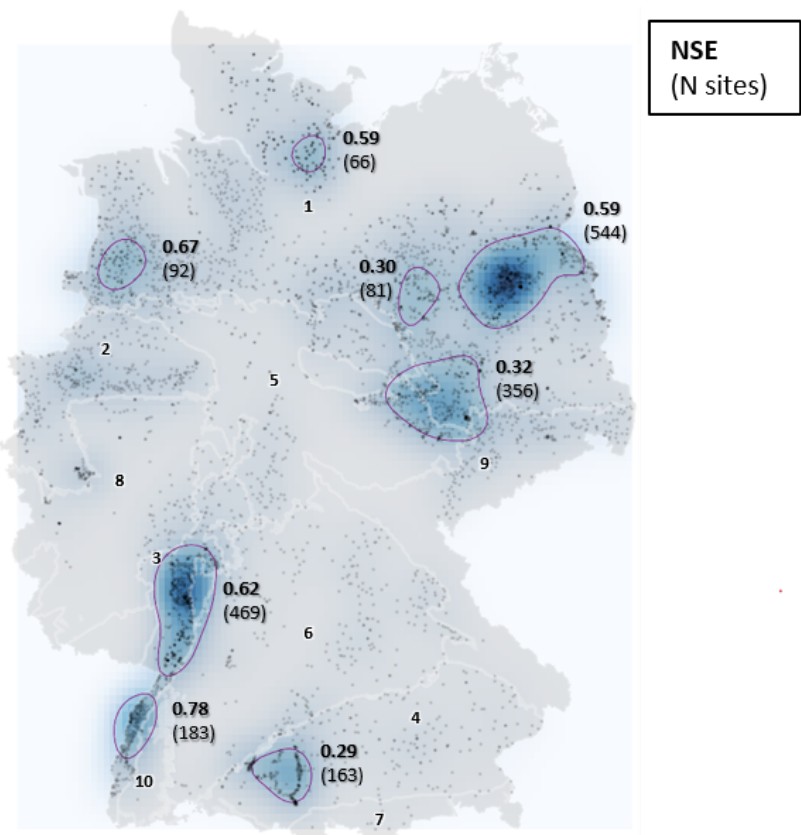

**Figure 5.** KDE of the density of monitoring wells. Areas with high data density are depicted in darker blue and encircled in purple. Median NSE values are reported in bold for each area with high data density. Dots represent the 5,288 monitoring wells. The white lines and numbers from 1-10 represent the major hydrogeological districts. Legend: *1) North and Central German Unconsolidated Rock District, 2) Rhenish-Westphalian Lowland, 3) Upper Rhine Graben with Mainz Basin and North Hessian Tertiary, 4) Alpine Foreland, 5) Central German Fault-block Land, 6) West and South German Scarplands and Fault-block Land, 7) Alps, 8) West and Central German Basement, 9) Southeast German Basement, 10) Southwest German Basement.*



### 3.3 Model performance comparison across static features and time series features

To understand under which conditions the ML models perform best and to assess whether they capture influential factors affecting groundwater level dynamics, an analysis on the correlation between model performance and both numeric static features and time series features of the groundwater hydrographs was carried out. The correlations were largely similar for the
245 N-HiTS and TFT model variants, hence results for the N-HiTS model provided with static and dynamic features for the 12-week prediction are reported in the text, if not denoted differently (Supplement Table 3). Similarly, regarding the performance distribution across the different static feature categories, the results for the best performing model, N-HiTS provided with static and dynamic features, are reported.

No strong correlations between the static numeric features and the NSE values for the 12 week prediction were found. Most
250 notable correlations were a positive correlation between the TWI and the NSE ($\rho = 0.22$) and negative correlations between elevation ($\rho = -0.27$) and the Landform Shannon Index ($\rho = -0.23$) with the NSE. For the time series features, the highest correlations were obtained for the seasonal behaviour (Table 3). Here, a positive correlation of $\rho = 0.28$ was obtained ($\rho = 0.34$ for N-HiTS (purely dynamic)). Other notable correlations were a negative correlation between NSE and the flashiness of the groundwater hydrographs ($\rho = -0.21$), and a positive correlation with the range ratio ($\rho = 0.17$). A weak correlation was
255 found between the NSE values and the linear trend as well as the NSE and the time series length ($\rho <= 0.03$). All reported correlations were statistically significantly different from zero (Table 3). Distributions of the time series features are provided in the Supplement (Figure B3). A wide variation in model performance was observed across the different static feature categories (Supplement Figure B4, Table B3). The highest median NSE values were achieved for monitoring wells situated in porous aquifers (0.52), at sites with moderate to high permeability coefficients of > 1E-5 to 1E-3 m/s (0.56), within closed deciduous
broadleaf forests (0.60), and in peatlands (0.66). 83 % of the wells were located in porous aquifers, and 68 % were positioned in areas with moderate to high permeability. Only a small proportion of wells were located within deciduous forests (4 %), primarily in the Upper Rhine Graben, or in peatlands (5 %), mostly in northern Germany (Supplement Figure B6). For the different major hydrogeological districts of Germany, the best results for the 12-week predictions were obtained in the North and Central German Unconsolidated Rock District, the Upper Rhine Graben and the Southwest German Basement (Median
NSE between 0.56 and 0.7). In these districts were either many wells located, they had a high density of wells, or they were located near districts with a high density of wells.




**Table 3.** Spearman correlations between the NSE values for the 12-week prediction horizon per monitoring well and the numeric static input features as well as the time series features. Correlations between NSE values for the purely dynamic model variants and the static numeric features were omitted, because these features were not an input to these models. Significance codes: '***' 0, '**' $<= 0.01$, '*' $<= 0.05$, ' ' $> 0.05$

| $\rho_{Spearman}$ | NSE TFT | NSE TFT (purely dynamic) | NSE N-HiTS | NSE N-HiTS (purely dynamic) |
|---|---|---|---|---|
| **Static Numeric Feature** | | | | |
| Elevation | -0.24*** | | -0.27*** | |
| EUMOHP_DSD1 | 0.09** | | 0.1** | |
| EUMOHP_LP1 | 0.1** | | 0.1** | |
| EUMOHP_SD1 | 0.14** | | 0.14** | |
| Groundwater recharge | 0.14** | | 0.1** | |
| Shannon Index Landforms | -0.23*** | | -0.23*** | |
| TWI | 0.18*** | | 0.22*** | |
| **Time Series Feature** | | | | |
| Seasonal behaviour | 0.27** | 0.30** | 0.28** | 0.34** |
| SDdiff (Flashiness) | -0.22*** | -0.23*** | -0.21*** | -0.19*** |
| Range ratio | 0.1** | 0.09** | 0.17*** | 0.21*** |
| Amplitude | -0.12** | -0.1** | -0.13** | -0.12** |
| Skew | -0.09** | -0.07** | -0.11** | -0.11** |
| Linear trend (slope) | 0.009* | 0.03** | -0.02** | -0.02** |
| Time series length | 0.01* | -0.002 | 0.02** | 0.03** |





### 3.4 Feature importance and attention of the TFT Models

The feature importance values of the TFT models derived from the variable selection networks were dependent on the model weight initialisations and showed a relatively high spread for most of the input features (Figure 6). The feature importance ranking was dominated for the dynamic input variables by the groundwater level. Other important dynamic features were precipitation, temperature and humidity, though for some initialisations the day of the year was also important (Figure 6, Supplement Figure B8). Among the static inputs, the standard deviation of the groundwater level was the most important input feature, followed by the TWI, the aquifer type, and the Landform Shannon Index.

The most important past time steps, according to the attention scores, were often at the beginning of the input sequences (52 weeks, i.e. the week a year ago) and recent time points. For the one-week prediction this was often the last time point and for the 12-week prediction this was often 12 weeks before the prediction (Supplement Figure B9, Figure B10).

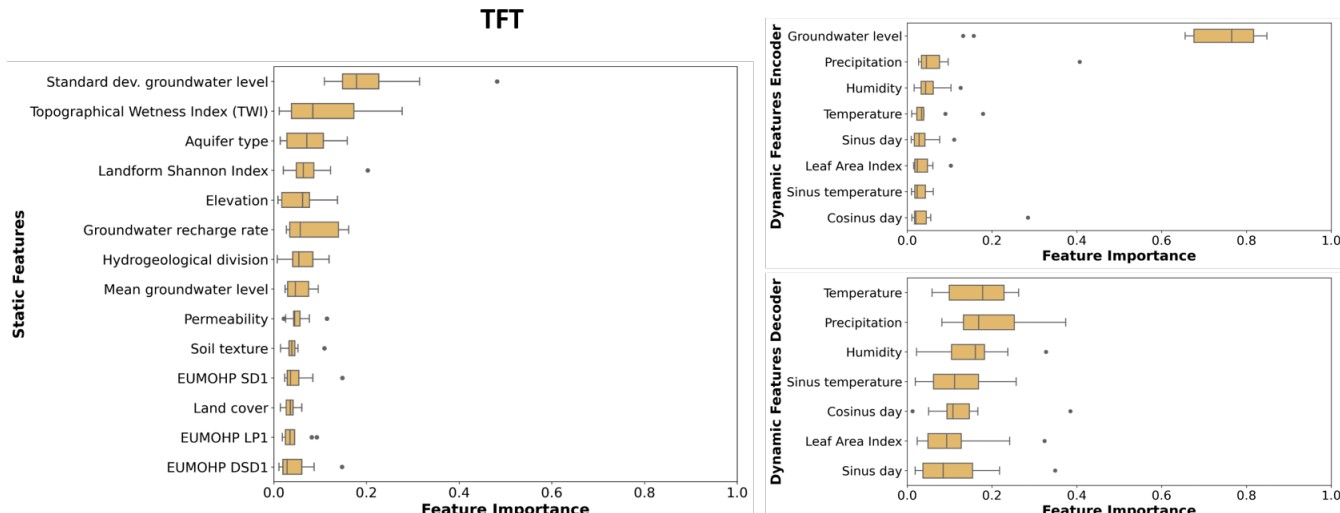

**Figure 6.** Variable importance of the TFT model trained with all static and dynamic input features. Each boxplot shows ten importance values based on the different weight initialisations. Variables are ranked based on their median importance. Results for the other TFT model variants are shown in the Supplement (Figure B8).



## 4   Discussion

### 4.1   General and Spatial Model Performance

The results for the 12-week prediction demonstrate that the tested global ML approaches can provide good predictions for
a large number of monitoring wells distributed over a broad region with diverse hydrogeological properties. Thereby, the N-
HiTS architecture outperformed the TFT architecture by achieving a median NSE of 0.5 across all 5,288 monitoring wells.
Both models showed high prediction accuracy's for a considerable number of wells when an NSE of at least 0.7 is considered
to be indicative of high performance (cf. Wunsch et al. (2022a)). Precisely, the N-HiTS model provided with static and dynamic
features achieved an NSE of at least 0.7 for 1163 monitoring wells (approximately 22 % of all monitoring wells) and the TFT
model achieved an NSE of at least 0.7 for 489 monitoring wells (9 % of all monitoring wells). Overall, the N-HiTS model
showed a good level of model performance, given the high amount of monitoring wells included without preselection and the
heterogeneous hydrogeological situation of the study area, comprising ten different major hydrogeological districts and 35
hydrogeological regions (BGR and SGD, 2019).

Two previous studies have predicted groundwater levels for up to 118 monitoring wells distributed across Germany using
single-well models based on Convolutional Neural Networks (CNNs) (Wunsch et al., 2022a) and a global LSTM architecture
(Heudorfer et al., 2024). 82 of these 118 monitoring wells were also used in our study. Nevertheless, direct comparisons to
our study should be made cautiously due to differences in input features and experimental design. The single-well models
solely used meteorological input features (Temperature and precipitation), while the LSTM approach included static features;
however, in both studies, groundwater level was not used as an input feature. The wells in these studies were preselected on
the basis that their groundwater dynamics were primarily influenced by climatic processes and could be accurately predicted
in the past. In addition, length of training, validation and test period were different compared to our study and Heudorfer et al.
(2024) also used partly different static input features. The single-well models achieved an NSE of at least 0.7 (median NSE =
0.81) and the global LSTM approach a median NSE of 0.82 for the one-week prediction horizon. For 82 of the 118 monitoring
wells, our best model, the N-HiTS model provided with static and dynamic features, achieved a NSE of 0.93 for the one-week
prediction horizon and a NSE of 0.62 for the 12-week prediction horizon. Thus, the N-HiTS model can compete with the
other approaches and is suitable for the purpose of short-term groundwater level prediction. However, N-HiTS in its current
implementation requires the target feature as input feature, and is for this task inferior to the single-well CNNs or the global
LSTM.

The N-HiTS model produced narrower prediction intervals based on the Interval Score, with the purely dynamic model
achieving narrower prediction intervals than the model provided with static and dynamic features for the 12 week prediction
horizon. Example hydrographs included in the supplement illustrate these results (Figure B1). This finding implies that when
the models have access to static features, they generalize or cluster across similar static feature levels, encountering a wide range
of groundwater dynamics. Consequently, prediction intervals may be wider than in purely dynamic models, which estimate
the 0.1 and 0.9 quantiles only based on historical groundwater levels and climatic data. This result indicates that the static
features currently lack sufficient "distinguishing potential" and may generalize across wells that are actually environmentally



distinct. One possible reason for this could be uncertainties in the static input features, which are discussed further in section subsection 4.3. Moreover, adding other relevant features, such as lithostratigraphic data at the monitoring well (which were unavailable for this study) could mitigate this issue.

The results further suggest that areas with a higher data density tend to have a higher prediction accuracy. Poor model
performances in high data density regions may be explained by specific regional and anthropogenic factors, e.g., former and recent activities of lignite mining in the southeast of the North and Central German Unconsolidated Rock District (Northwest Saxony) possibly lead to difficulties in predicting groundwater level fluctuations despite the high density of monitoring wells. For instance, Schroeter and Gläßer (2011) report a decline of the groundwater table of up to 70 m during the period of active mining in some parts of this area. Moreover, comparatively larger errors at certain locations and the erratic spatial pattern
of model performance suggest that there are influencing factors not covered by the input features used, these could include anthropogenic influences such as water withdrawals. Filtering out monitoring wells with strong anthropogenic influences is not trivial for such a large study area, as most wells will exhibit some form of "unnatural" influence. Future studies should aim for identifying time series features that indicate strong anthropogenic influences, which can then be used to filter or cluster for monitoring wells suitable for groundwater level prediction when information on water abstraction is not available.

**4.2 Hydrogeological and Environmental Drivers of Model Performance**

The correlation analysis of the NSE values with static features and time series features revealed weak or no correlations. The analysis of model performance across the different static feature categories indicated a few environmental conditions were a better performance could be achieved, but did not show pronounced differences.

The highest identified correlations suggest that the ML models performance tended to improve when groundwater dynamics
followed the expected seasonality, at monitoring wells with higher TWI values, at monitoring wells located at lower elevations, at monitoring wells that were surrounded by fewer or a less diverse set of landforms, and when groundwater hydrographs had a lower flashiness. The first four factors are potentially characterising monitoring wells in lowlands, as TWI values are higher at lower elevations ($\rho = -0.43$), probably with a low depth to groundwater, enabling a distinct seasonality. These findings align with Gomez et al. (2024) who also found that model accuracy improved in areas with higher TWI and decreased in
hilly regions. Examples of individual monitoring wells with particularly poor 12-week predictions where the expected seasonality was distorted (i.e. highest groundwater level in March, lowest in September) and with high flashiness, are shown in the Supplement (Figure B7). The analysis of model performance across the categorical static features implied a better model performance in hydrogeologically relatively homogeneous areas (i.e. major hydrogeological districts) where many wells are located. Moreover, model performance tended to be better in porous aquifers and at sites with medium to moderate perme-
ability coefficients. This finding aligns with theoretical expectations that in porous aquifers and at moderate permeabilities groundwater flow is relatively slow and uniform and thus more predictable than in other hydrogeological systems such as karst and fractured aquifers, which exhibit heterogenous subsurface conditions with high flow velocities and heterogeneous groundwater dynamics (Bakalowicz, 2005; Hermans et al., 2023). Furthermore, this finding is also likely related to the high number of training examples for these conditions. Two additional environmental conditions with higher NSE values stood out:




monitoring wells located in closed forests with deciduous broadleaf trees and monitoring wells located in peatlands. Wells
in closed forests with deciduous broadleaf trees were primarily located in an area with high data density, the Upper Rhine
Graben, suggesting that data availability rather than the presence of closed forests explain the good performance. In peatlands
hydraulic conductivity is generally assumed to be low in the deeper peat layers (Kværner and Snilsberg, 2011). Additionally,
high water tables may prevail because of limited lateral flow in areas with a low topographic gradient like Northern Germany
were most of the monitoring wells in peatlands are located (Oosterwoud et al., 2017). Thus, groundwater dynamics were more
controlled and potentially mainly contingent on the climatic signal, which can explain the good predictive performance. In fact,
the monitoring wells located in peatlands exhibited also relatively high values for the time series feature seasonal behaviour
(median 0.59).

     Overall, some hydrogeological and environmental conditions could be identified under which a good model performance
could be achieved, in line with the understanding of the hydrogeological system, but the heterogeneous results also show the
complexity of the data at hand.

## 4.3    The Role of Static Features in Global Machine Learning Models

Despite the relatively high feature importance scores for some static features, they contributed to improved predictive perfor-
mance in only a small number of monitoring wells, suggesting limited utility for generalisation across data points with similar
static similar levels. Previous studies demonstrated a stronger increase in predictive performance of global ML models when
incorporating static features (Heudorfer et al., 2024; Li et al., 2022). However, these studies were conducted for a much smaller
number of monitoring wells and the authors suggest that their models used the static features primarily as unique identifiers
(Heudorfer et al., 2024; Li et al., 2022). One possible explanation that the addition of static features yielded only to a small
increase in the overall model performance is that the used ML models may have primarily learned from the dynamic input fea-
tures. In contrast to the cited studies, the N-HiTS and TFT models were provided with the target variable (groundwater level)
as input feature, which was unexpectedly, the most important input feature based on the feature importance of the TFT. It is
important to note that by using a validation set and various techniques such as dropout and early stopping to avoid overfitting,
the models were prevented from simply replicating historical groundwater levels. Moreover, in comparison with the model
using solely dynamic input features, the model variants incorporating additional static features were partly "confused" by these
information. While for few monitoring wells an improvement in model performance with the addition of static features could
be achieved, e.g., for N-HiTS for the 12 week prediction 5.4% of monitoring wells with an NSE of 0.5 saw an increase in model
performance of 0.1 NSE or greater, for a small portion of monitoring wells the model performance declined. To be precise, 1.9
% of monitoring wells with an NSE of at least 0.5, exhibited a decrease in model performance of 0.1 NSE or greater. Possibly,
the wells exhibiting a decline in performance had groundwater dynamics that differed strongly from other monitoring wells
with similar static feature values.

     Further reasons why the models may have been unable to extract meaningful relationships between the groundwater level
and the static features may be due to their inherent uncertainties, which make it challenging for the models to use them to
generalise to wells with similar static feature levels. These uncertainties stem from 1) the fact that the best spatial aggregation





that accurately reflects the static input features and their relation to the groundwater level depends on the specific feature

and monitoring location and may not be localised to the pixel level (1 km buffer) at the surface, and 2) the regionalised nature of these features, wherein values are derived by extrapolating from discrete points into broader spatial contexts. In order to improve future groundwater level predictions, it could be beneficial to utilise higher resolution maps or combine two-dimensional maps with time series data (e.g., Addimando et al. (2022)), as well as to test different spatial aggregations of static features. The features employed in this study are predominantly describing surface or near-surface conditions (e.g.,

soil texture), since they are easily accessible and available for a large area. Groundwater flow dynamics are strongly affected by subsurface factors and the underlying complexity of the hydrogeology. Thus, the incorporation of features that describe subsurface properties, such as the depth and thickness of the aquifer in the vicinity of the monitoring wells, could enhance the predictive performance. However, at present, information on such features is often not available extensively.

Lastly, the marginal improvement in model performance for the N-HiTS model with static features might be attributed to

390 differences in the data fusion of static and dynamic input features compared to the TFT architecture. N-HiTS uses an additional stack composed of fully connected layers for exogenous variables (FCNN), while the TFT uses GRN encoders that provide context for the temporal processing.

## 4.4 Feature importance and important past time steps

The TFT offers an intrinsic feature importance method through the VSNs, providing insights into which features are relevant

for prediction. This makes this architecture attractive, as no post-analysis on feature importance was required, such as Shapley values or permutation importance. Nevertheless, this study demonstrated that the results should be interpreted with caution, as some static features exhibited relatively high importance values (standard deviation of groundwater level, TWI), yet did not enhance the overall predictive performance of the TFT. Moreover, the feature importance values of the TFT had a high dependency on the model weight initialisation which underpins the necessity to run several intialisations when using this

approach. The observed variability in feature importance for the features day of the year, the temperature, and the sinus of the temperature can be attributed to the fact that these features provide the model with information on the yearly seasonality. If one of them becomes important, the importance of the other features tends to diminish.

The TFT architecture also offers insights into the most important past time steps through the attention scores. The attention scores reflected the seasonality in the data and highlighted its importance as well as the importance of the immediate past for

groundwater level prediction. That attention scores from a TFT model mainly reflect seasonality in hydrological or hydrogeological data was also observed in a study on streamflow prediction (Koya and Roy, 2023). Future studies may explore using longer sequence lengths to capture longer range dependencies, wherein attention layers can reveal the relevance of these time points, though such experiments likely come with higher computational costs.




## 5 Conclusion

In this study, two recently developed global ML architectures, TFT and N-HiTS, were applied to predict groundwater levels for up to 12 weeks for 5,288 monitoring wells distributed across Germany. The results demonstrated that these architectures are suitable for seasonal groundwater level prediction for thousands of monitoring wells, enabling good predictions across a large area with only one model, whereby the N-HiTS model showed better predictive performance for our dataset. While overall predictive performance was good the models failed to accurately predict groundwater levels for a considerable number of wells
(about 15 % with an NSE below 0 with the best performing model), which highlights the hydrogeological complexity of the study area and the possible influence of anthropogenic factors such as water withdrawals, which could not be captured by the input features. To address this issue, future research should aim at filtering out monitoring wells with strong anthropogenic influences, for example, based on suitable time series features, or, if possible, include features that describe anthropogenic factors such as water abstractions.

In contrast to our expectations, the addition of static features into the models did result in improved performance only for a small number of monitoring wells. Regarding the overall performance across all 5,288 monitoring wells, the static features marginally improved the overall performance of the N-HiTS model, but not for the TFT model. Future studies could examine the potential benefits of alternative representations of static features, such as two-dimensional maps or more detailed information on subsurface processes, in order to more accurately reflect the true conditions at and around a monitoring well.
Additionally, testing and using different buffer sizes for data extraction around the monitoring wells could lead to further improvements in the use of static features.

The correlations between model performance and different static features, as well as time series features, were found to be low. This may again be attributed to the complex hydrogeological conditions across Germany, as well as potential anthropogenic influences. Nevertheless, the results of the correlation analysis implied that more accurate predictions can be made in
lowland areas with less complex surrounding landscapes and groundwater hydrographs exhibiting a seasonal pattern. Moreover, comparisons of model performance across the categorical static features pointed out that a slightly better performance can be obtained in porous aquifers, areas with medium to moderate permeability coefficients, and in peatlands. Areas with a high data density appear to facilitate more accurate predictions, which aligns with theoretical expectations that high-capacity ML models benefit from a large number of training examples with a high data diversity (Nearing et al., 2021; Kratzert et al.,
2024).

The feature importance obtained from the TFT showed a relatively high variability across the ten initialisations. This should be taken into account when utilising the model interpretability insights provided by the TFT, or alternatively, model-agnostic techniques should be employed. Nevertheless, the results indicated that the TFT mainly learned from the provided groundwater levels and the standard deviation of the groundwater levels. Other important dynamic features were precipitation, humidity,
and temperature, while a notable important static feature was the TWI.

In conclusion, the state-of-the-art global ML architectures TFT and N-HiTS represent valuable additions to the thus far established LSTM architecture for groundwater level prediction. The N-HiTS architecture demonstrated superior predictive





capabilities compared to the TFT architecture based on the predictive performance on the compiled data set. This makes it a powerful ML architecture for groundwater level predictions in large, hydrogeologically complex areas.

*Code and data availability.*  The code necessary to reproduce our results is available on GitHub: https://github.com/KunzstBGR/global-groundwater-models-main. The TFT and N-HiTS architectures were implemented and trained with Pytorch forecasting (Version 1.0.0) and Pytorch lightning (Version 2.0.4), which are opensource deep learning modules based on the Pytorch framework (Paszke et al., 2019). A complete list of all the used python modules is available at the GitHub repository (requirements_GGWM.txt file). All groundwater level data are available free of charge from the respective local authorities upon request.

*Author contributions.*  SK: conceptualisation, data curation, investigation, formal analysis, software, writing. AS: conceptualisation, data curation, software, writing, MW: conceptualisation, data curation, visualisation, writing (review & editing), MN: data curation, writing (review & editing), TC: software, writing (review & editing), FB: conceptualisation, resources, methodology, writing (review & editing), SB: conceptualisation, funding acquisition, resources, writing (review & editing)

*Competing interests.*  The contact author has declared that none of the authors has any competing interests.



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



**Appendix A:  Additional information on training data and model evaluation**

**A1    Length of training period**

**Table A1.** Descriptive statistics on the length of the training period.

|  | Length training period (y) |
| --- | --- |
| Nr. monitoring wells | 5,288 |
| Mean | 16.42 |
| Std | 4.68 |
| Min | 6 |
| 25% | 13 |
| 50% | 20 |
| 75% | 20 |
| Max | 20 |

**A2    Description Root Mean Bias Error and Interval Score**

As additional error metrics the relative mean bias error (rMBE) and the Interval Score for evaluating probabilistic forecasts are
reported. The rMBE is defined as the mean deviance divided by a constant, for which the standard deviation of the groundwater
levels is chosen. Negative rMBE values indicate underestimation, positive values indicate overestimation and a value close to
zero can indicate a good estimate. Nevertheless, a value of zero does not necessarily indicate a perfect estimate, since under
-and overestimation can cancel each other out. Hence, the rMBE is only sensitive to biases having a dominant direction. It is
defined as follows:

$$rMBE = \frac{1}{sd(\mathbf{y})} \frac{\sum_{t=1}^{T}(\hat{y}_t - y_t)}{T} \tag{A1}$$

The Interval Score (Equation A2) combines two properties of an prediction interval: (1) the sharpness, which refers to the
concentration of the predictive distribution and is a property of the prediction only and (2) the calibration, which denotes how
much the true value is outside of the predicted interval. In the equation $t$ denotes a time step in the time series, $T$ denotes the
total number of time steps, $y_t$ denotes the observed value of the target variable at time step $t$, $\hat{y}_t$ denotes the estimated value
at time step $t$, $\hat{y}_t^{\top}$ and $\hat{y}_t^{\perp}$ denote the upper and lower bound of the predicted distribution at time step $t$. Smaller values of the
Interval Score indicate a better estimate and a narrower prediction interval. A normalised version of the Interval Score was
used, obtained by dividing the Interval Score by the standard deviation of the true values.



$$rIS = \frac{1}{sd(\mathbf{y})} \frac{\sum_{t=1}^{T} \left( \overbrace{(\hat{y}_t^{\top} - \hat{y}_t^{\perp})}^{\text{sharpness}} + \overbrace{\frac{2}{\alpha} \left( max(\hat{y}_t^{\perp} - y_t, 0) + max(y_t - \hat{y}_t^{\top}, 0) \right)}^{\text{calibration}} \right)}{T} \tag{A2}$$





**Appendix B:  Model performance and model interpretability**

**B1    Model performance comparison across prediction horizons**

**Table B1.** Median NSE values for the prediction of groundwater levels of the 5,288 wells for each horizon. Reported values are based on the median NSE of the 10 initialisations for each well. P-values refer to the difference between the model variants provided with all input features compared to the purely dynamic model (One-sided Mann-Whitney-U test).

| Horizon | Median NSE TFT | Median NSE TFT (purely dynamic) | P-value TFT | Median NSE N-HiTS | Median NSE N-HiTS (purely dynamic) | P-value N-HiTS |
|---|---|---|---|---|---|---|
| 1 | 0.91 | 0.91 | 3.22e-03 | 0.92 | 0.92 | 8.89e-02 |
| 2 | 0.85 | 0.85 | 5.71e-02 | 0.87 | 0.86 | 3.91e-02 |
| 3 | 0.80 | 0.80 | 1.29e-02 | 0.82 | 0.82 | 1.83e-02 |
| 4 | 0.75 | 0.74 | 5.76e-03 | 0.79 | 0.78 | 4.84e-03 |
| 5 | 0.69 | 0.69 | 2.56e-03 | 0.74 | 0.73 | 3.01e-03 |
| 6 | 0.64 | 0.63 | 2.53e-03 | 0.70 | 0.68 | 2.30e-03 |
| 7 | 0.59 | 0.57 | 3.69e-03 | 0.66 | 0.64 | 6.66e-04 |
| 8 | 0.54 | 0.53 | 2.00e-02 | 0.62 | 0.61 | 5.20e-04 |
| 9 | 0.48 | 0.48 | 2.34e-02 | 0.59 | 0.57 | 2.05e-05 |
| 10 | 0.44 | 0.43 | 3.51e-02 | 0.55 | 0.53 | 5.84e-05 |
| 11 | 0.39 | 0.39 | 2.59e-01 | 0.53 | 0.51 | 3.18e-05 |
| 12 | 0.34 | 0.35 | 4.21e-01 | 0.50 | 0.47 | 7.42e-06 |





**Table B2.** Median Interval Scores for TFT and N-HiTS models for each horizon. Reported values are based on the median interval score of the models provided with all input features compared to the purely dynamic model. P-values refer to the difference between the model variants and the purely dynamic model (Two-sided Mann-Whitney-U test).

| Horizon | Median Interval Score TFT | Median Interval Score TFT (purely dynamic) | P-value TFT | Median Interval Score N-HiTS | Median Interval Score N-HiTS (purely dynamic) | P-value N-HiTS |
|---|---|---|---|---|---|---|
| 1 | 1.54 | 1.62 | 8.634e-24 | 1.19 | 1.26 | 1.659e-07 |
| 2 | 1.98 | 2.14 | 4.029e-43 | 1.86 | 1.70 | 1.698e-25 |
| 3 | 2.51 | 3.15 | 1.192e-286 | 2.25 | 2.19 | 7.403e-03 |
| 4 | 2.86 | 3.01 | 7.466e-28 | 2.65 | 2.44 | 1.694e-32 |
| 5 | 3.29 | 3.44 | 2.533e-15 | 3.01 | 2.88 | 9.202e-15 |
| 6 | 3.65 | 4.08 | 3.570e-111 | 3.16 | 3.16 | 1.982e-01 |
| 7 | 3.72 | 4.07 | 4.571e-63 | 3.62 | 3.41 | 9.869e-24 |
| 8 | 4.13 | 4.56 | 2.129e-109 | 3.65 | 3.69 | 6.766e-01 |
| 9 | 4.81 | 4.53 | 2.223e-59 | 3.82 | 3.96 | 3.939e-09 |
| 10 | 4.44 | 4.82 | 2.233e-68 | 3.84 | 3.90 | 1.114e-01 |
| 11 | 4.53 | 4.61 | 3.054e-03 | 3.99 | 3.80 | 1.023e-17 |
| 12 | 4.45 | 4.41 | 4.126e-08 | 3.89 | 3.68 | 2.131e-17 |





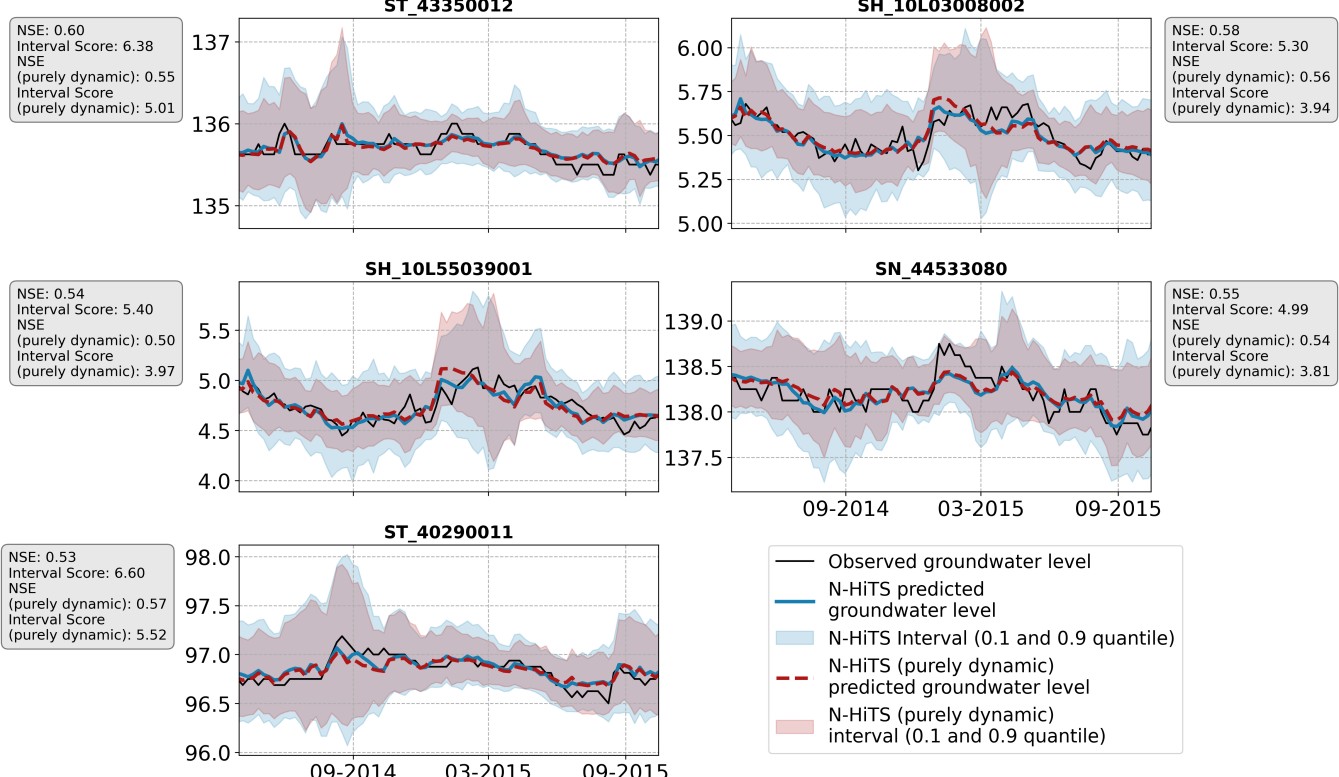

**Figure B1.** Five example hydrographs with narrower prediction intervals obtained with the N-HiTS (purely dynamic) model than the N-HiTS model provided with static and dynamic features.



## B2 Spatial patterns of model performance

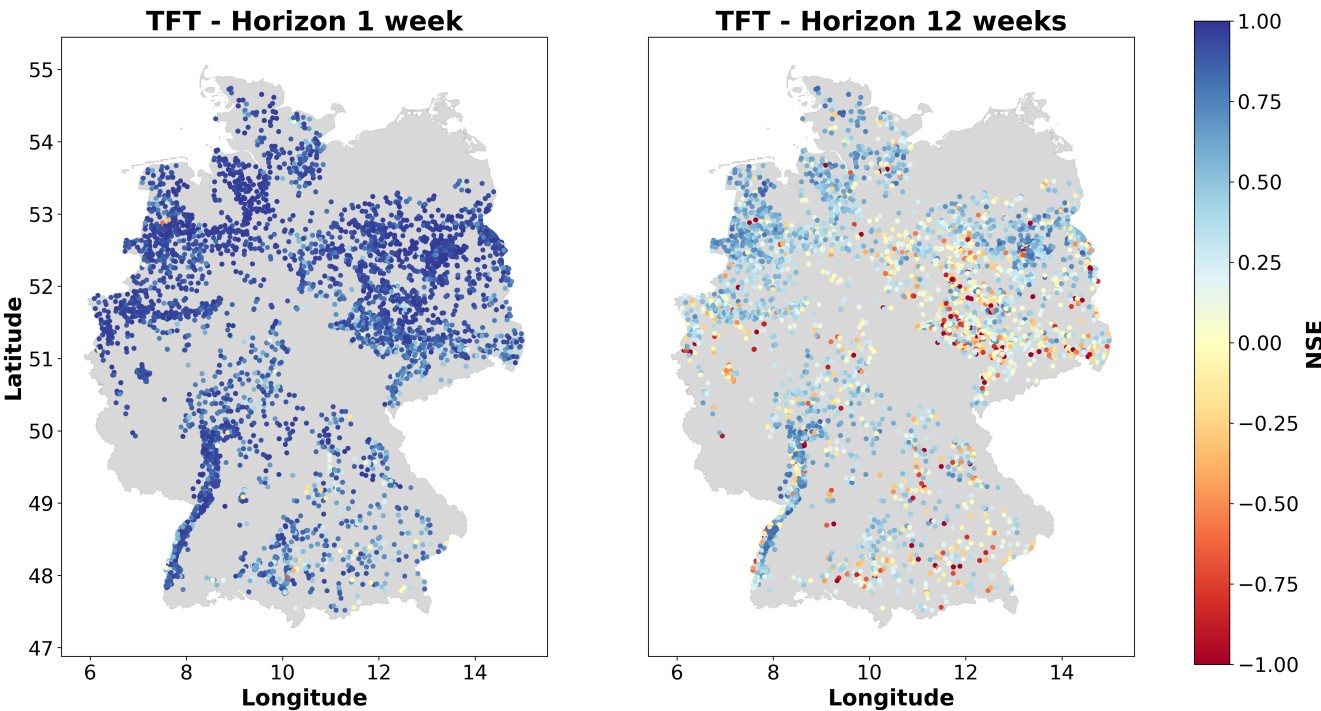

**Figure B2.** Spatial distribution of the NSE values for TFT model provided with static and dynamic features across Germany for the one-week prediction and 12-week prediction horizon.





## B3  Hydrogeological and Environmental Drivers of Model Performance

**Figure B3.** Distributions of the calculated time series feature values for the 5,288 monitoring wells.




Table B3: Median NSE values for different static features for all model variants and prediction horizon 12 weeks. *n* is the number of wells in each major hydrogeological district. The legend for the major hydrogeological districts is shown in Figure B5.

| Static feature | *n* | Median NSE N-HiTS | Median NSE N-HiTS (purely dynamic) | Median NSE TFT | Median NSE TFT (purely dynamic) |
|---|---|---|---|---|---|
| **Aquifer Type** | | | | | |
| Fractured and karstified aquifers | 185 | 0.45 | 0.44 | 0.33 | 0.33 |
| Fractured and porous aquifers | 95 | 0.37 | 0.36 | 0.27 | 0.26 |
| Fractured aquifers | 612 | 0.39 | 0.36 | 0.27 | 0.27 |
| Porous aquifers | 4395 | 0.52 | 0.49 | 0.36 | 0.36 |
| **Land cover** | | | | | |
| Closed forest, deciduous broad leaf | 187 | 0.6 | 0.56 | 0.39 | 0.43 |
| Closed forest, evergreen needle leaf | 463 | 0.55 | 0.48 | 0.35 | 0.36 |
| Closed forest, mixed | 129 | 0.55 | 0.48 | 0.37 | 0.42 |
| Cultivated and managed vegetation/ agriculture | 3611 | 0.51 | 0.49 | 0.36 | 0.35 |
| Herbaceous vegetation | 65 | 0.34 | 0.32 | 0.23 | 0.2 |
| Open forest, mixed | 3 | 0.42 | 0.57 | 0.42 | 0.52 |
| Open forest, unknown | 15 | 0.06 | -0.22 | -0.29 | -0.17 |
| Urban/built up | 792 | 0.4 | 0.37 | 0.29 | 0.31 |
| **Major hydrogeological district** | | | | | |
| 1 | 2564 | 0.56 | 0.54 | 0.4 | 0.4 |
| 2 | 387 | 0.47 | 0.46 | 0.36 | 0.35 |
| 3 | 807 | 0.64 | 0.62 | 0.54 | 0.52 |
| 4 | 422 | 0.3 | 0.24 | 0.17 | 0.14 |
| 5 | 441 | 0.32 | 0.31 | 0.17 | 0.19 |
| 6 | 265 | 0.49 | 0.44 | 0.35 | 0.36 |
| 7 | 12 | 0.26 | 0.26 | 0.14 | 0.14 |
| 8 | 164 | 0.35 | 0.34 | 0.22 | 0.29 |
| 9 | 211 | 0.22 | 0.18 | 0.08 | 0.11 |
| 10 | 15 | 0.7 | 0.68 | 0.58 | 0.56 |





**Permeability coefficient**

| | | | | | |
|---|---|---|---|---|---|
| High (>1E-3 - 1E-2) | 788 | 0.37 | 0.33 | 0.22 | 0.21 |
| Highly variable | 110 | 0.29 | 0.26 | 0.24 | 0.21 |
| Low to extremely low (<1E-5) | 520 | 0.38 | 0.36 | 0.26 | 0.25 |
| Medium to moderate (>1E-5 - 1E-3) | 3600 | 0.56 | 0.53 | 0.4 | 0.4 |
| Moderate to low (>1E-6 - 1E-4) | 265 | 0.33 | 0.29 | 0.24 | 0.23 |

**Soil texture**

| | | | | | |
|---|---|---|---|---|---|
| Clay loams | 592 | 0.47 | 0.48 | 0.38 | 0.39 |
| Clay silts | 876 | 0.38 | 0.36 | 0.25 | 0.28 |
| Loams | 127 | 0.32 | 0.3 | 0.17 | 0.08 |
| Loamy sands | 1376 | 0.57 | 0.54 | 0.42 | 0.41 |
| Loamy silts | 287 | 0.38 | 0.34 | 0.2 | 0.24 |
| Open cast mining sites | 7 | 0.19 | 0.27 | 0.14 | 0.19 |
| Peatlands | 286 | 0.66 | 0.63 | 0.46 | 0.44 |
| Pure sands | 807 | 0.6 | 0.58 | 0.43 | 0.43 |
| Sandy loams | 558 | 0.44 | 0.38 | 0.29 | 0.27 |
| Silty clays | 54 | 0.5 | 0.46 | 0.36 | 0.29 |
| Silty sands | 19 | 0.29 | 0.38 | 0.24 | 0.18 |
| Urban settlements | 287 | 0.35 | 0.33 | 0.26 | 0.32 |







**Figure B4.** Barplots showing the median NSE per static feature category for the N-HiTS model provided with static and dynamic features for the 12 week prediction horizon. The categories with the highest values are highlighted. The sample size is given behind each bar. The dotted vertical line indicates the overall NSE for this model variant (0.5). Legend for the major hydrogeological districts in Figure B5.



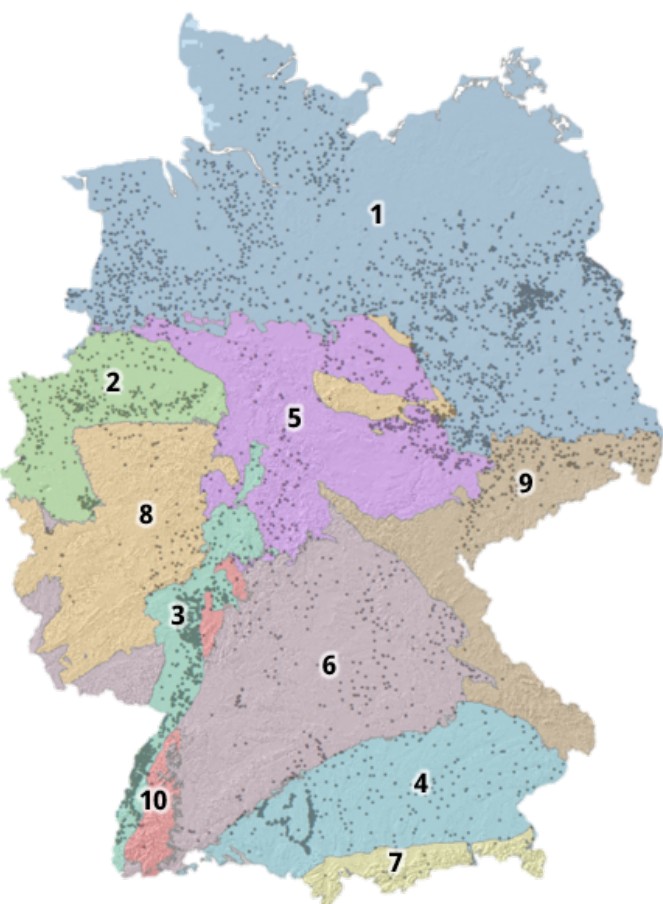

**Figure B5.** The major hydrogeological districts for Germany. Dots represent the 5,288 monitoring wells. Legend: *1) North and Central German Unconsolidated Rock District, 2) Rhenish-Westphalian Lowland, 3) Upper Rhine Graben with Mainz Basin and North Hessian Tertiary, 4) Alpine Foreland, 5) Central German Fault-block Land, 6) West and South German Scarplands and Fault-block Land, 7) Alps, 8) West and Central German Basement, 9) Southeast German Basement, 10) Southwest German Basement.*





**Figure B6.** Locations of the monitoring wells with the highest median NSE values for N-HiTS model provided with static and dynamic features for the 12 week prediction horizon for monitoring wells located in porous aquifers, at sites with medium permeabilities, closed forest with deciduous broad leaf, and in peatlands.





## B4   Predictions for individual wells

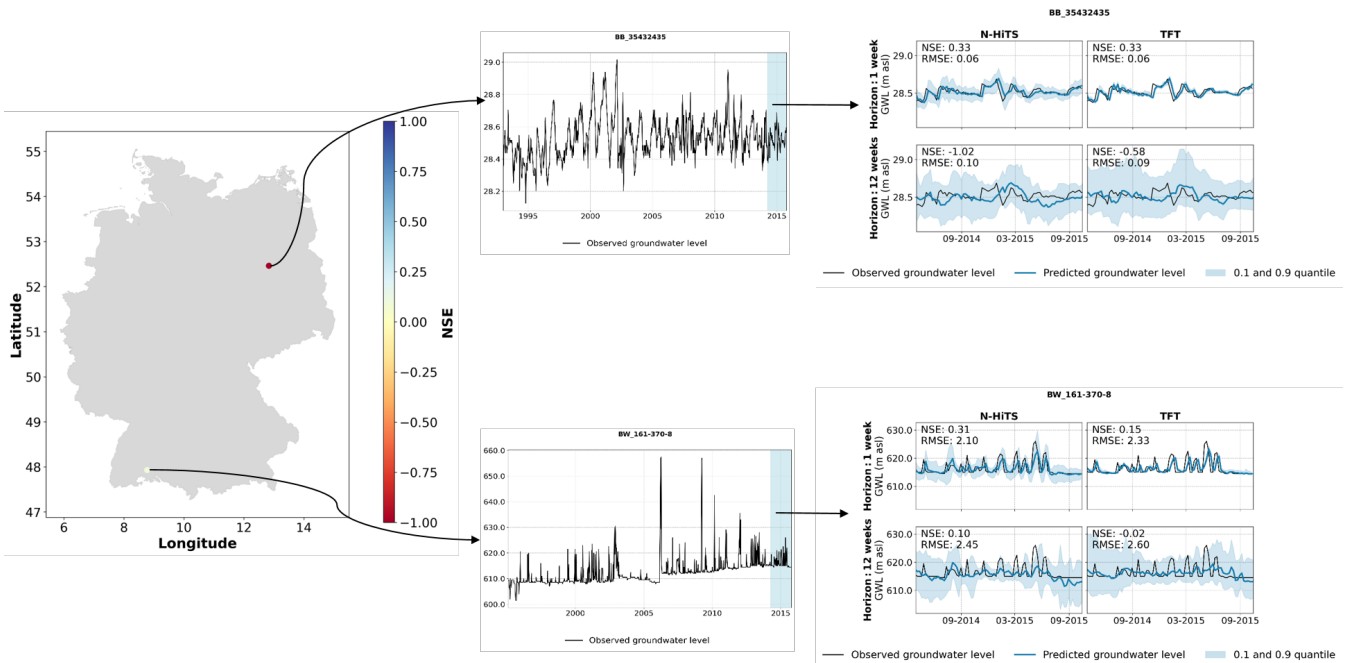

**Figure B7.** Examples of two monitoring wells with groundwater hydrographs with a distorted seasonality (BB_35432435) and a high flashiness (BW_161-370-8). The middle panel shows the hydrographs for the entire period of training, validation and test (blue shaded area marks the test period). The right panel shows the predictions for the N-HiTS and TFT model provided with static and dynamic input features for the test period.




## 605  B5  Feature importance of the TFT Models

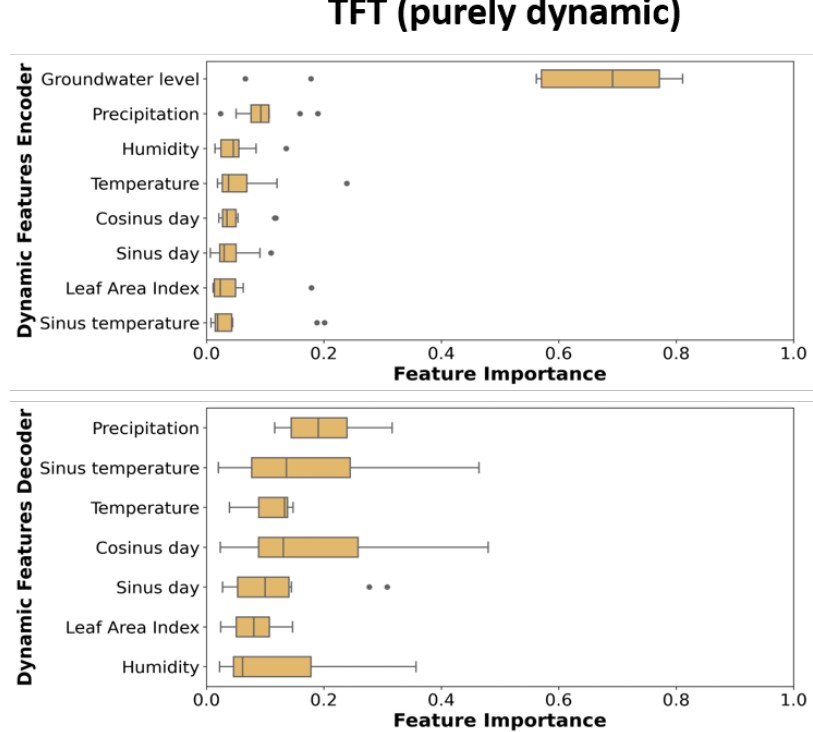

**Figure B8.** Variable importance's of the TFT (purely dynamic) model. Each boxplot shows ten importance values based on the different weight initialisations. Variables are ranked based on their median importance.





## B6    Attention curves

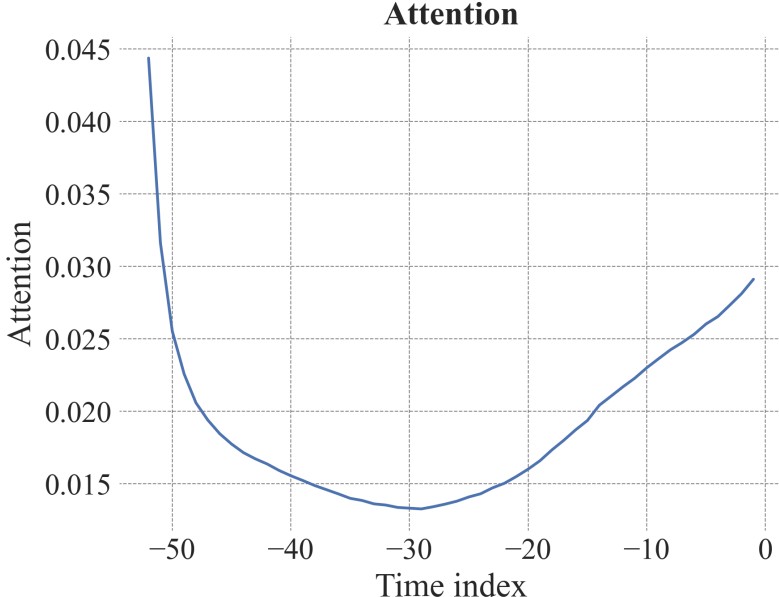

**Figure B9.** Attention curve for the one-week prediction of the TFT provided with static and dynamic input variables highlighting important past time steps. Time index is in weeks.





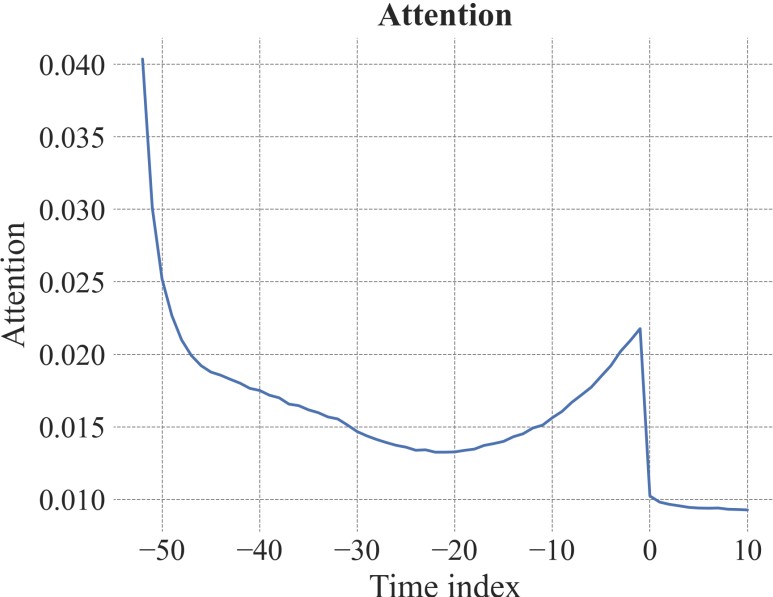

**Figure B10.** Attention curve for the 12-week prediction of the TFT provided with static and dynamic input variables highlighting important past time steps. Time index is in weeks.