# Peer review of "Towards a Global Spatial Machine Learning Model for Seasonal Groundwater Level Predictions in Germany"

_EGUsphere, 2024_

## Author Comment (AC1)

**Review 1**

This manuscript evaluates the performance of two machine learning models in predicting groundwater (GW) levels across a dataset of ~5000 wells in Germany. The study examines the influence of both dynamic and static input features on the accuracy of GW level predictions and seeks to enhance the understanding of hydrogeological systems.

The objectives, methodology, results, and discussion are clear, well-structured, and thoroughly explained. The study aligns well with the scope of the journal, and HESS readers would benefit from and appreciate its findings. In my opinion, the manuscript is close to its final form. However, I would like to raise the following points for consideration:

We thank Reviewer 1 for the positive assessment of our manuscript. Below we address the suggestions (marked in blue).

- The manuscript specifies particular values for hyperparameters (e.g., dropout rate, batch size, etc.). Are these values based on specific rules or conventions? Did you test alternative values? While this may not significantly affect the overall conclusions, I believe it would be helpful to clarify this for the reader.

Thank you for your suggestion. We initially selected hyperparameter (HP) values based on empirical heuristics recommended by domain experts, aiming to reduce overfitting and minimize training time. As noted in line 160 ff., the final values for dropout rate and batch size were chosen to achieve these goals. The decision to set the number of training epochs to ten was based on the convergence of the training and validation loss observed during preliminary testing. In particular, models based on the N-HiTS architecture would converge before the 10th epoch.

We will modify section 2.3 (lines 160 ff.) accordingly:

"All model variants were trained with ten different random seeds to account for the stochasticity in the initialisation of model weights. Large batch sizes were used (TFT: 4096, N-HiTS: 1024) to avoid overfitting and to accelerate the training. The risk of overfitting was further reduced by the application of early stopping on the validation loss, a dropout rate of 0.2, and learning rate scheduling using stochastic weight averaging after the second epoch (Izmailov et al., 2019). Thus, the selected hyperparameter values were based on empirical heuristics recommended by domain experts, aiming to reduce overfitting and minimize training time. All model variants were trained for a maximum of ten epochs, a duration sufficient to ensure model convergence. In many cases, training terminated earlier due to the implementation of the early stopping criteria."

- Is there a specific reason for setting the prediction horizon to a maximum of 12 weeks?

The aim of our study was to provide seasonal groundwater level predictions. The different seasons are known for their substantial impact on groundwater recharge, and thus on groundwater levels. Accordingly, we selected a prediction horizon of 12 weeks, equivalent to approximately three months, as an appropriate timespan for reflecting seasonal patterns. Furthermore, we observed a decrease in model performance with longer horizons as shown in figure 3. A 12-week prediction horizon allowed us to maintain acceptable predictive performance across the more than 5,000 monitoring wells.

Figures B9 and B10 indicate that attention is higher one year before the prediction than at times closer to it. Could you elaborate on why this happens?

We interpret the results in figures B9 and B10 as likely related to the autocorrelation function of many of the observed groundwater level time series. Based on the intrinsic feature importance of the TFT, we know that the most important feature is the historical groundwater level. For the one-week prediction, the groundwater level from the corresponding week one year prior to the prediction is likely the most influential information for the TFT models, as reflected in the attention scores. This is likely due to the seasonality observed in many groundwater monitoring wells. Accordingly, for the 12-week prediction, the groundwater levels from the week one year prior and 12 weeks before the prediction are the most important time steps. To support our interpretation, we will include the autocorrelation for the 52$^{nd}$ week as well as the 12$^{th}$ week of each groundwater hydrograph in the Supplement as additional explanation. We will also add a reference in the description of figures B9 and B10 to the autocorrelation.

- In figure B10, why are attention values not zero in the interval of 0–10 weeks? Does this imply that the algorithm is somehow using inputs from these time steps? I suggest including a diagram to illustrate how inputs and outputs operate in the ML algorithms (e.g., similar to Figure 1 in Kratzert et al., 2018). This would help clarify which specific information is being utilized and when.

We thank the Reviewer for the suggestion to clarify how input features operate in the models. We will add a figure in the method section.
Both architectures follow an encoder-decoder structure. The encoder creates a latent representation of the input features, while the decoder uses this representation to generate predictions. The historical groundwater levels, historical climatic features and the static features are processed in the encoder. In the decoder, the climatic features for the prediction horizon (so-called future knowns) and the static features are used, while groundwater levels are not used. The non-zero attention values observed during the 12-week prediction likely reflect the information the Temporal Fusion Transformer (TFT) extracts from the future known features during these time steps.

---

## Author Comment (AC2)

**Review 3**

The submission by Kunz et al. presents the development and application of a machine learning model for groundwater level prediction in Germany. The models are referred to as "global" since they are trained against a multitude of wells simultaneously. The study entails several novel aspects which make the submission highly relevant for publication in HESS: 1) the applied models have not previously been applied in the groundwater domain and go beyond the state of the art, 2) such a large number of monitoring wells with time series data has not been used for model development before, and 3) the thorough investigation of the effect of static features in the models.

We thank Reviewer 3 for the positive assessment of our manuscript. Below we address the suggestions (marked in blue).

I only have a few comments that I wish to see addressed before publication:

1. **Introduction**: The cited literature in the introduction could be diversified. Here are two suggested references that could be included:
   - Collenteur, R. A., Haaf, E., Bakker, M., Liesch, T., Wunsch, A., Soonthornrangsan, J., ... & Meysami, R. (2024). Data-driven modelling of hydraulic-head time series: results and lessons learned from the 2022 Groundwater Time Series Modelling Challenge. Hydrology and Earth System Sciences, 28(23), 5193-5208.
   - Chidepudi, S. K. R., Massei, N., Jardani, A., & Henriot, A. (2024). Groundwater level reconstruction using long-term climate reanalysis data and deep neural networks. Journal of Hydrology: Regional Studies, 51, 101632

Thank you for the suggested literature. We will add the references in the introduction where studies for groundwater level prediction are discussed (line 4 ff.).

2. **Section 2.1.1**: This section is missing information on the temporal resolution of the data. What is the frequency of the measurements, and were the measurements aggregated in time?

The temporal resolution of the groundwater measurements was weekly after the data preprocessing steps. This resolution of the groundwater levels is reported in table 1. Prior to the preprocessing, some of the raw groundwater level observations were in part on monthly resolution. Those groundwater level observations have been upsampled via linear interpolation before we obtained the data.

3. **Section 2.3**: Please clarify if the models are run in an autoregressive manner, simulating one timestep at a time (i.e., prediction at t1 is added to the dynamic inputs to predict t2), or if a sequence for the entire forecast horizon is outputted directly.

Thank you for the suggestion. The models generate predictions for entire sequences at once, i.e. seq2seq prediction. In our case, the model predicts a sequence of groundwater level values for 12 weeks. We will clarify this in section 2.3 (line 159 ff.):

"Both ML architectures used in this study are designed for sequence-to-sequence predictions. During training, the models processed an input sequence autoregressively and predicted an output sequence of groundwater levels. For each time step, a look-back window (i.e. sequence length) of 52 weeks for the dynamic features was used to represent one annual cycle. Groundwater levels were predicted for 12 weeks. During the 12 week prediction the model has access to the exogenous dynamic features, but not to the groundwater level."

4.      **Section 2.4**: Please clarify how the prediction intervals have been utilized. Were three separate models trained for the 0.1, 0.5, and 0.9 quantiles?

The prediction intervals were obtained with one model. The model learns to predict different parts of the conditional distribution simultaneously. Computing the loss for multiple quantiles results in a multi-output model, with output dimensions (horizon x quantiles). During training the loss is reduced via averaging.

5.      **Discussion**: Given the data presented in this paper, I was hoping the authors would attempt predictions at ungauged wells. Currently, groundwater level observations are used both in the dynamic and static features, making predictions at ungauged wells impossible with the existing model setup. I encourage the authors to add a discussion section outlining a path towards predicting groundwater levels at ungauged wells. This could be supported with an additional model experiment that excludes observed groundwater level data from the input features and is based on a spatial hold-out of monitoring wells for model testing. Even a poor test performance of such a spatio-temporal holdout experiment would be relevant to publish to underline the need for future research. To my knowledge such an experiment has not been published yet

For seasonal or short term prediction all available information, i.e. both endogenous (lag features or historical measurements) and exogenous inputs, are typically used to improve the predictive performance. Thereby, the historical measurements of the target feature serve as a starting point for the prediction. For long-term predictions such as decadal predictions or when the aim is to predict wells a strategy without the target feature as input feature (exogenous-only) is necessary, to which the reviewer is referring to. However, this was not the aim in our study.

Nevertheless, in preliminary experiments, we have evaluated the importance of historical groundwater levels as input by making predictions with the Temporal Fusion Transformer (TFT) without the historical groundwater level as an input. The model's performance deteriorated with an estimated NSE of 0.37 (with static features) and 0.22 (purely dynamic) for the one-week prediction, and an NSE of 0.08 (with static features) and -0.04 (purely dynamic) for the 12 week prediction. We will include these analyses in the supplement and mention the results in the manuscript. As the performance of the TFT model across all 5,288 monitoring wells was considerably worse without the groundwater levels as input feature, we decided not to pursue further analyses in this regard. It is important to note that our study includes a large number of monitoring wells with varying degrees of predictability, including many located in hydrogeological complex areas or influenced by anthropogenic effects. This contrasts with other studies that often focus on wells with inherently high predictive capacity (e.g. wells predominantly influenced by climatic factors), leading to higher NSE values. Nevertheless, there are still 884 wells in our study where the TFT model provided with static features achieved an NSE >= 0.5 even without groundwater levels as an input feature.

Moreover, we believe that the topic of prediction in ungauged wells is beyond the scope of this study. The main focus of our study is on seasonal predictions, where the inclusion of historical measurements of the target feature as input is standard practice to enhance the models performance. Removing this critical feature would not align with our study's objectives, which aim to evaluate the utility of advanced machine learning architectures in a realistic operational setting. Heudorfer et al. (2024) have carried out analyses on the predictive capabilities on ungauged wells with a global LSTM model.

**References**

Heudorfer, B., Liesch, T., & Broda, S. (2024). On the challenges of global entity-aware deep

learning models for groundwater level prediction. *Hydrology and Earth System Sciences*,

*28*(3), 525–543. https://doi.org/10.5194/hess-28-525-2024

---

## Author Comment (AC3)

**Review 2**

General comments :

The paper is related to the needs of groundwater modeling using exlclusively machine learning approaches. Whereas this field of research is not complety new, the use of two specific model architecture (TFT and N-HiTS) has never been published in hydrogeology which represents a novelty. In addition, the study combines both static and dynamic features in the dataset. Static features have been carefully selected to represents hydrogeological properties, or representing links with groundwater levels that hydrogeologist can easily figure out.

The very large number of wells for training worth also to be noted as a novelty, and represents a clear advance in the field of hydrogeology, as it suggest that this model architecture could be suitable for even larger datasets. On top of this, confrontation with hydrogeologists expertise is made, which is both hardly found in litterature and very insightfull.

The overall structure is clear and leads the reader to a detailed understanding of  methods, workflow, results and main outcomes.

We thank Abel Henriot for his positive assessment of our manuscript. Below we address his suggestions (marked in blue). We have taken the liberty to number his remarks to facilitate cross-referencing between them.

1) Objectivs or motivation are slightly less clearly explained. And I think a 2 to 3 sentences paragraph at the end of the introduction that exposes more clearly all the motivations could be usefull. From my understanding :

* more efficient models for enlarged dataset compared to previous study (Heudorfer et.al.)

* more capabilities to handle static and dynamic features

* need to enhance the overall performance (NSE of 0.8 in Heudorfer et.al.)

* need to understand/evaluate impact of introduction static features

* capability of models to stick to plausible hydrogeological concepts

Thank you very much for this suggestion. We will modify the second-to-last paragraph in the introduction accordingly (line 65 ff.) : "This study seeks to expand upon this research by using a larger number of monitoring wells and exploring more complex ML architecture. The primary objectives are: 1) to introduce models based on modern ML architectures for groundwater modelling in addition to the established LSTM; 2) to evaluate the models performance with a substantially larger dataset than those used in prior research; 3) to evaluate the importance of input features, with a specific focus on static features, in influencing the prediction accuracy; and (4) to assess the capability of these models to align with hydrogeological system understanding. We hypothesise that the large amount of time series data used in combination with static input features will help to achieve a good predictive performance. ..."

Results part has been kept short, with help of complementary material, and only the key points have been reported. This is one of the good point of this paper.

Discussion  highlight very crucial consideration, in particular in the field of generalization capacities for this models, and information availabilty for static features.

Thank you very much.

2) I have a minor issue with the fact that waterlevel itself has been included in the input features. While this is a common way to proceed and is perfectly grounded, comparison with competing models that did not use this feature is made. I'm questionning myself on the reason of this choice and the overall impact of it.

For seasonal or short term prediction all available information, i.e. both endogenous (lag features or historical measurements) and exogenous inputs, are typically used to improve the predictive performance. Including the historical measurements of the target feature, such as groundwater levels, is crucial because it provides direct information on the system's current state and is particularly helpful when the aim is to generate operational short-term predictions. Thereby, the historical measurements of the target feature serve as a starting point for the prediction. For long-term predictions such as decadal predictions or when the aim is to predict in areas without historical groundwater level measurements (i.e., prediction in ungauged basins) a strategy without the target feature as input feature (exogenous-only) is necessary. However, this was not the aim in our study.

In preliminary experiments, we have evaluated the importance of historical groundwater levels as input by making predictions with the Temporal Fusion Transformer (TFT) without the historical groundwater level as an input. The performance of both model variants deteriorated with an estimated NSE of 0.37 (with static features) and 0.22 (purely dynamic) for the one-week prediction, and an NSE of 0.08 (with static features) and -0.04 (purely dynamic) for the 12 week prediction. Thus, we opted to include the groundwater level as an input feature. Furthermore, the current implementation of the N-HiTS architecture in the python module pytorch-forecasting which we have used requires the target feature to be an input feature.

Generally, we agree with the reviewer that the comparison to Heudorfer et al. (2024) and Wunsch et al. (2022) is not directly possible and we have addressed differences in input features, experimental design and monitoring well selection in the discussion (line 290 ff.). The comparison was made because both studies made predictions for monitoring wells in Germany, partly for wells included in our study.

3) Last, most of the references are for German works, and I missed a more general overview of ML for groundwater when it comes to deals to overall performance of the models.

We will add comparisons to other studies that used ML approaches to predict groundwater levels, for example to Collenteur et al. (2024) or Chidepudi et al. (2024) (preprint). Nevertheless, so far, we think there's a lack of international studies applying modern machine learning methods for groundwater level prediction, particularly in a global setting. Through our results, we hope to demonstrate the value of investing in large datasets for groundwater modeling. Furthermore, the comparisons to other studies conducted in Germany were made because of the partial overlap of monitoring wells.

All in all, this paper is, to me, an important step for the community, and the code hase been made available which should foster competitors to try for their dataset.

Thank you very much!

Specific comments

4) 2.1.1 : time step of measurement are not reported, and no mention to re-scaling where needed (time resolution harmonisation)

The temporal resolution of the groundwater measurements was weekly after the data preprocessing steps. This resolution of the groundwater levels is reported in table 1. Prior to the preprocessing, some of the raw groundwater level observations were in part on monthly resolution. Those groundwater level observations have been upsampled via linear interpolation before we obtained the data. See also our response to remark 6).

Also, the aggregation of the dynamic features (e.g. for temperature, daily values are aggregated to weekly values with the mean) is reported in table 1. If the reviewer thinks this could be better placed in sections 2.1.1 and 2.1.2 we are open to do so.

5) line 88 : 50 times the average : on what basis is this figure based ? trial and error ? user expertise ? it's surprisingly far from often seen methods such as '3 standard deviation', or 1.5 IQR, z-scores, etc.

We agree with the reviewer that this is not a standard method. This threshold was chosen after extensive manual testing and represents a conservative strategy, ensuring that only extreme outliers, which are clearly implausible, are removed. This conservative approach minimizes the risk of removing legitimate data while ensuring data quality

6) line 89 : 4 weeks : this could be challenged as hydaulic head variability could be low or high within this time-period depending on the influences under which the well reacts

While we agree with the reviewer, this decision was necessary since some of the raw groundwater level observations had in part monthly values. In order not to lose too much information, those were upsampled via linear interpolation to obtain weekly values.

7) line 91 : could be great to justify that the time series are in steady state and that 1996-2010 ; 2010-2013 and 2013-2016 exhibits no major change in time series characteristics, so that the train/validation/test datasets share similar (if not equals) characteristics

Thank you for the suggestion. We will include a statistical summary of the mean, variance, minimum, and maximum for each groundwater level time series during the training, validation, and test periods, and add this information to the Supplement.

8) end of § 2.1.1. Hard to figure out, whith no prior knowledge on Germany if the few missing data would lead to ignore some specific climatic, geological, hydrogeological contexts. I'll encourage the authors to add a short sentence in this direction.

We thank the reviewer for this suggestion. With information for over 5000 monitoring wells we cover all major relevant hydrogeological and climatic conditions of Germany, even though some federal states were not represented. Accordingly, we will add the following sentences in the method section (line 99 ff.):

"Nevertheless, all major hydrogeological and climatic conditions of Germany are represented in our dataset. For example, in the hydrogeological district that includes Thuringia, common aquifer types include fractured, fractured and karstified, as well as fractured and porous aquifers. Monitoring wells located in these aquifer types are also present in other regions covered in our dataset (e.g., over 612 monitoring wells are located in fractured aquifers, SI Table B3). "

2.1.2 dynamic features :

9) Snow, wind, solar radiation has not be used. I would expect snow cover and snow melt to play a role for some years, and some parts of Germany. ANd solar radiation and wind in the evapotranspiration process. Even if I do believe that not all meteorological features can be considered, it could be insightfull for the reader to know if this variables are available in HYRAS, and why they have not been included in the dataset (redudancy for example ?).

The selection of the meteorological input features was based on theoretical expectations regarding the key drivers of groundwater dynamics, as well as on features that have been successfully applied in previous studies (see Heudorfer et al. (2024), Wunsch et al. (2021, 2022)). While we agree with the reviewer that incorporating additional climatic features could potentially enhance groundwater level prediction models, much of this information is currently unavailable. For example, data on wind and snow are not provided by HYRAS.

10) line 101 'which have a strong influence on groundwater' sounds to be not really grounded on any scientific aruguments. I would suggest to rephrase: (meteorological datasets that are regularly used in modeling (reservoirs ou distributed models), or in similar studies (add some international references))

Thank you for this suggestion. We will rephrase the sentence (line 101 ff.) to: "Further dynamic input features were precipitation (mm), relative humidity (%), and temperature (°C), which have been successfully used in previous modelling studies (see Heudorfer et al. (2024) and citations therein) and should theoretically exert a strong influence on groundwater levels. These meteorological dynamic features were extracted from HYRAS 5.0 (Razafimaharo et al. 2020)."

11) 102,103 - units are written in parenthesis, maybe easier or better to be integrated in table 1, and for all variables

We agree with the reviewer and will integrate the units into table 1.

12) 105 : grid size is not given, so it becomes unclear of what is the impact of the weighted average step (averaging several pixels i.e. the 1km buffer includes several (how many ?) smaller pixels, or downsampling i.e. 1 pixel is >> than the 1km buffer)

The grid size for precipitation in HYRAS 5 is 1 KM x 1 KM. For temperature and humidity the grid size is 5 KM x 5 KM. We will add this information to the manuscript (line 105 ff.). That means the buffer typically includes only a few other pixels.

13) line 109, 110 : "The meteorological input features and the LAI were extracted within a 1 km buffer around the groundwater wells. Thereby, a weighted average was calculated based on the area covered by the pixels within the buffer." this should be challenge or at least discussed in the perspective of my previous remark, one can think that averaging could not allways be the best method

Given the relatively coarse grid size (see our response to comment 12), we believe that averaging is a reasonable aggregation method and that using the median, for example, would be unlikely to affect the results much. Generally, we agree with the reviewer that the aggregation within the buffer as well as the buffer size influences values of the dynamic features for each monitoring well. We address this point in subsection 4.3.

To the best of our knowledge, systematic comparisons to determine the optimal aggregation method and buffer size for groundwater level prediction have not been conducted and are beyond the scope of our study.

14) line 111 I think the last sentence "Dynamic features were divided [...] groundwater time series" would benefit to be moved in the 2.3 paragraph (experimental design and training)

We thank the reviewer for this suggestion. We will move this sentence as well as the information on splitting the groundwater levels into training, validation and test dataset given into section 2.1.1 to section 2.3.

15) line 113 : "The static features used in the study are environmental characteristics from the domains hydrogeology, soil, topography and land cover (see Table 2)." The sentence sounds wired to me. Maybe the verb 'are' could be changed to 'covers'

Thank you, we will adapt the sentence accordingly.

16) table 1 : could it be possible to describe which variables are considered as 'meteorological input features' (see lines 102, 103) ?

The meteorological input features are described in table 1. They are: temperature, the sinus of the temperature, precipitation, and the relative humidity. We will modify lines 102 ff. according to our answer to remark 10.

17) line 125 'vanilla LSMT'. considering this paper could be read by hydrogeologist with no priori knowledge in computer science, i'll suggest to avoid this term that sounds as technical jargon /technical language

We agree with the suggestion and will omit the word "vanilla" and just write "LSTM" in line 125.

§2.2

18) The models architectures are well described even if remains unclear if some steps have been specifically designed for the study or are included (or part of) the overall TFT models (e.g. "The importance of each feature is then represented by the average of the variable selection weights over all time steps.")

I would suggest to split the 2.2.1 into two parts : 1-overall description of TFT and 2-implementation with the frame of the study

There has been no specific implementation of the used model architectures for this study. The models were utilized in their standard forms as described in the original publications by Lim et al. (2020) and Challu et al. (2022).

19) However I still think that the motivation for new architecture is not justified enough, eventhough one can almost understand some of them :

more efficient models for enlarge dataset compared to previous study (Heudorfer et.al.)

more capabilities to handle static and dynamic features

need to enhance the overall performance (NSE of 0.8 in Heudorfer et.al.)

We thank the reviewer for this suggestion and refer to remark 1), where we will explicitly justify the use of new architectures at the end of the discussion.

Strategy for model evaluation is clever and robust, with consideration for hydrogeological context.

Thank you very much.

§2.3 - experimental design and training

20.1) I would expect to find here the global strategy for train/validation/test and not in 'data (§2.1.3)'.

We agree with the reviewer and will move the information of the train/validation/test split to section 2.3 (see response to remark 14).

20.2)  Also, a complementary strategy could have been to split on wells themselves, leaving aside a given proportion of wells and their related data. I think this could improve the paper to explain why this has not been used.

Out-of-sample cross-validation in the spatial domain was not conducted in this study because such an approach is primarily useful to evaluate the generalization ability of models to unseen conditions (see Heudorfer et al. (2023, 2024)). While we acknowledge that this is an important and interesting area of research, we consider it beyond the scope of our study. Our focus was on assessing the predictive capabilities of modern ML architectures, the importance of input features, and the extent to which the model's results align with hydrogeological system understanding. See also our answer to remark 2.

21) The assesment of the impact of static feature is obviously a good strategy, but this is not justify in the previous section of the paper. In my opinion this also one of the objectiv of this paper, and if so, should be expressed as so. I suggest to add a sentence in the last part of introduction that more clearly explain that the evaluation of impact of static features

We hope that we have addressed this point with our answer to remark 1 and the suggested changes therein.

22.1) line 159 : Groundwater levels were predicted from one up to 12 weeks. For every time step, [...]

This is unclear to me. Would it means the models predicts several sequences of growing lenghts ranging from 1 to 12 weeks ? or it is a sequence [52 weeks] to value [horizon = 1 week] procedure with recursion to achieve the 12 weeks desired horizon ?

I think it would be preferable to clearly highlight : the prediction horizon, the time step, and the sequence to sequence or sequence to value prediction strategy.

Thank you for the suggestion. The models predict sequences in one go, i.e. seq2seq prediction. In our case the model predicts a sequence of groundwater level values for 12 weeks. We will clarify this in section 2.3 (line 159 ff.):

"Both ML architectures used in this study are designed for sequence-to-sequence predictions. During training, the models processed an input sequence autoregressively and predicted an output sequence of groundwater levels. For each time step, a look-back window (i.e. sequence length) of 52 weeks for the dynamic features was used to represent one annual cycle. Groundwater levels were predicted for 12 weeks."

22.2) Are the dynamic variables (covariates) known/given to the model for each predicted  time steps ? i.e. water level is predicted from past 52 weeks + of water level, rainfall, etc.. + 2 past weeks of rainfall, evaporation, etc... in case of a 2 weeks ahead prediction ?

The dynamic covariates are known to the model in the prediction phase (so-called future knowns). Both used ML architectures follow an encoder-decoder structure. The encoder creates a latent representation of the input features, while the decoder uses this representation to generate predictions. The historical groundwater levels, historical climatic features and the

static features are processed in the encoder. In the decoder, the future knowns and the static features are used, while groundwater levels are not used.

We will add a figure in the method section to clarify which features are used by the model in the encoder and decoder.

23) lines 160-167 a short discussion on the hyperparameters values could be interesting : how this 10 epochs, dropout rate of 0.2 have been choosen ? DId you made any test on this values to track the gain/loss on the overall model performance ?

We initially selected hyperparameters (HPs) based on empirical heuristics recommended by domain experts, aiming to reduce overfitting and minimize training time. As noted in line 160 ff., the final values for dropout rate and batch size were chosen to achieve these goals. The decision to set the number of training epochs to ten was based on the convergence of the training and validation loss observed during preliminary testing. Especially, models based on the N-HiTS architecture would converge before the 10th epoch.

We will modify section 2.3 (lines 160 ff.) accordingly:

"All model variants were trained with ten different random seeds to account for the stochasticity in the initialisation of model weights. Large batch sizes were used (TFT: 4096, N-HiTS: 1024) to avoid overfitting and to accelerate the training. The risk of overfitting was further reduced by the application of early stopping on the validation loss, a dropout rate of 0.2, and learning rate scheduling using stochastic weight averaging after the second epoch (Izmailov et al., 2019). Thus, the selected hyperparameter values were based on empirical heuristics recommended by domain experts, aiming to reduce overfitting and minimize training time. All model variants were trained for a maximum of ten epochs, a duration sufficient to ensure model convergence. In many cases, training terminated earlier due to the implementation of the early stopping criteria."

24) "The quantile loss was chosen because it is more robust towards outliers than for example the root mean squared

error (RMSE), e.g., caused by extreme precipitation events." : or a reference or word to express that it's your expertise, or after trial against other loss function.

We will modify the sentence: "The quantile loss was chosen because it has a linear relationship with the error magnitude, making it more robust towards outliers than metrics with a quadratic relationship, such as the mean squared error (MSE). Outliers may arise, e.g., from extreme precipitation                                                                                                events."

25) 194- Since the paper deals with two architecture (TFT and NHiTS), it's needed to say smth on both of them. Does NHiTS offers similar capabilities ?

An intrinsic feature importance is not implemented in the N-HiTS architecture. However, N-HiTS offers the visualisation of the decomposed basis function representing different temporal resolutions that contribute to the forecast. Theoretically, the decomposed basis functions could be aggregated for each hydrograph to obtain information on the shared temporal pattern between hydrographs. Since our study focuses on predictive performance, the influence of input features on the performance, as well as the hydrogeological system understanding captured by the model,  we consider such an analysis as beyond the scope of our study. Hence, we did only mention the TFT model regarding explainability.

If the reviewer wishes, we offer to add the following sentence (line 195 ff.): "The N-HiTS architecture offers the visualisation of the decomposed basis functions representing different

temporal resolutions for each hydrograph, but does not provide an intrinsic feature importance.
"

Results :

26) 198 - one-week prediction ... Depending on the dataset, this could be not completly relevant. There is no clue in the paper on the proportion of wells that exibit low frequency variations (i.e. inertial or very inertial). For such cases, even simple models (persistance, or even exponential smoothing) performs all ready very well

To provide a better evaluation of the predictive performance for the one-week prediction, we propose to incorporate a comparison to a naive forecast. We anticipate that the models we present in the paper will align with the naive forecast for inert hydrographs, but will be superior in cases where hydrographs exhibit a higher flashiness.

27) line 201 (TFT 0.34....) -> proposition to write (whereas median NSE is 0.34 for TFT...), otherwise hard to understand where the 81 % comes from.

We will follow the suggestion of the reviewer and will rewrite the sentence (line 201 ff.):

"The N-HiTS model provided with static and dynamic features achieved a median NSE of 0.5 across the 5,288 monitoring wells, whereas the TFT reached an median NSE of 0.34  (Figure 2A, Supplement Table B1), meaning that for approximately 81 % of the monitoring wells (4286 monitoring wells) N-HiTS achieved higher NSE values than the TFT."

28) 205 - term 'ground truth' if known from advanced user of ML could be hard to understand for other hydrogeologist. If ground truth here is the observed hydraulic head, it could be interesting to more clearly say it.

We agree with the reviewer and will replace "ground truth" with "observed groundwater levels".

29) fig 2. The only axis that do not share the same (xmin, xmax) is the 2B) bottom right, which makes the comparison harder.

We agree with the reviewer and will adapt the figure.

30) fig 3. is underrated. Very little is said on the basis of it. That's a shame because i find it really interesting : the decrease in performance appears almost linear for the interval of 1 - 12 weeks. It also support the fact that extremes prediction horizon 1 and 12 weeks only have been shown, and all the other ignored. SInce there is no sharp change in the performances, there is no prediction horizon at which the model performance is getting really worst and that is also in intersting point. TFT exhibits a difference compared to NHiTS : median for static + dynamic and purely dynamic are almost the same, while for NHiTS, there is a visible difference of about 0.2 point at 12 weeks.

We thank the reviewer for his suggestion and will provide additional description for figure 3. Based on the observed linear decrease in performance, we decided to show results only for the one and 12 week prediction horizons, because general patterns in performance difference are similar for the other prediction horizons. Moreover, we have addressed the differences in performance between the model variants trained with and without static features (see line 220 ff., Figure 2A).

We will include the following sentence in the manuscript (line 208 ff.): "Generally, model performances decreased approximately linearly across the prediction horizons, and the patterns observed for the one- and 12- prediction horizons are representative for the other prediction horizons (Figure 3)."

31) line 227-228 : again, at 1 week, I do agree that the models performances are high, but I would find interesting to mitigate this claim as it's higlhy plausible that any model and even simple one would have good performances.

We agree with the reviewer and will add a comparison with a naive forecast for the one-week prediction horizon.

32) "Poor performances with a median NSE below zero" -> is this for the 1 week horizon only ? If yes, a word like 'However' at the begining of the sentence would help to clarify.

Yes, this sentence refers to the one-week predictions. We will follow the suggestion of the reviewer.

33) line 230 - reference to regional terms (Upper Rhine Graben, Central German Unconsolidated Rock District, Alpine Foreland) are  confusing because they hav'nt been describre before, and no reference is made to the figure B5 in supplementary material. I would recommend to add a short § (1 to 2 sentences) in section 2-data to explain briefly the geology of Germany and make a reference to the B5 figure.

We agree with the reviewer that mentioning the hydrogeological districts there without reference in the method section is confusing for the reader. We will add a reference in section 2.4 where we describe the model evaluation (line 180 ff.):

"To investigate if certain hydrogeological conditions allow better predictions, the NSE values of each monitoring well were evaluated for a prediction horizon of 12 weeks across the categorical static features, including the large hydrogeological districts of Germany (Supplement Figure B5)"

Since the large hydrogeological districts are in detail described in the reference provided in Table 2 we refrain from explaining them in the manuscript. We will add this reference also to figure B5.

34) figure 5 - KDE density varies from 0 to 1, right ? Is so, maybe add it in the legend. Hydrogeological units are hard to read. Here again, a reference to the B5 figure would be helpfull, as I don't see any easy way to improve readibility (maybe try to thicken the white line, or use a medium grey ?).

We thank the reviewer for his suggestion and will improve the visibility of figure 5 and also add a legend

35) line 244 : the term correlation refers to the 'spearman correlation coeficient ?'

Yes. we will adapt the sentence accordingly to make this clear to the reader.

36) line 245-248 Why the 1 week horizon is no more in the race as in the begining of the paper ?

For the one-week prediction we only report performance, because the model is expected to primarily extrapolate from the provided historical groundwater level one week prior. In contrast, for the 12-week prediction, we expect the model to use information from the other dynamic and static features, i.e. that it is making predictions rather than merely extrapolating. Furthermore, the 12-week horizon aligns with the desired timeframe for operational future forecasting. Therefore, the error metrics and model performance for this horizon are of particular importance, as they directly reflect the model's utility in practical applications.

Discussion - 4.1

37) line 281. and NSE of 0.5 is not really high. This should be mentionned somewhere, for the reader to see that authors know there is a room for improvement.

In this line we refer to a high NSE of at least 0.7 based on Wunsch et al. (2022a)). We assume the reviewer refers to line 280.

To the best of our knowledge, there is no clear accepted distinction in hydrogeology regarding what constitutes a high or low NSE, other than an NSE below 0 denotes inferior performance. Such an assessment is contingent on the extent of the studied area, the complexity of the geology therein, and the number of monitoring wells included. Given the complex hydrogeological situation in Germany and the high number of monitoring wells included in the study, we consider an NSE of 0.5 to still represent good performance. This has been clarified in lines 284 and following. We have also identified potential ways to improve the performance (e.g., lines 320 ff., lines 375 ff.). Additionally, it is important to highlight that many previous studies primarily focus on monitoring wells with inherently high predictability, which leads to higher NSE values in their results. In contrast, our study deliberately includes a large number of wells, some of which are influenced by anthropogenic effects, such as groundwater abstraction or artificial recharge, which naturally reduces predictive performance. Therefore, error metrics and NSE values are not directly comparable across studies, as they are heavily influenced by the specific selection of wells and the underlying hydrogeological and anthropogenic conditions.

38.1) line : 291 "The single-well models solely used meteorological input features (Temperature and precipitation), while the LSTM approach included static features"

line : 294 "The wells in these studies were preselected on the basis that their groundwater dynamics were primarily influenced by climatic processes [...]"

line 301 "However, N-HiTS in its current implementation requires the target feature as input feature, and is for this task inferior to the single-well CNNs or the global LSTM."

and line 274 - "The most important past time steps, according to the attention scores, were often at the beginning of the input sequences (52 weeks, i.e. the week a year ago) and recent time points. " + the feature importance of the 'groundwater level' (figure 6) suggest that the vast majority of wells exhibits an annual regime with very little variations around this. This suggest that TFT of NHiTS are very capable of replicating the past patterns of groundwater heads (low flow in summer, high flow in winter), but not very capable to understand the transformation of rain into groundwater levels evolution (through /infiltration/recharge and possibly delay into the unsaturated zone, and up to the top of aquifer).

While we found that the models perform well for monitoring wells with an expected seasonal pattern, there are also instances where the models effectively capture unusual dynamics, such as hydrographs with a weak seasonality and extended phase of recession or a downward trend. We will include two such examples in Figure B7 to illustrate this and describe these examples in the manuscript. These examples are well BB_29519013 and ST_47360014:

[Figure]

BB_29519013

[Figure]

ST_47360014

38.2) This 4 parts of the paper makes a strong inconstancy. While the objectivs (from what I can guess) are probably to compete whith the Heudorfer implementation (LSTM) and do groundwater prediction, the case where the groundwater level itself is left aside appears not to be considered. In short : why a model without groundwater level as input feature has not be evaluated ? what happens when this feature is left aside on the overall model performance ? I did not found anywhare justification for this choice, and comparision with previous work makes this justification unavoidable.

Our objective is not to compete with the model of Heudorfer et al. (2024), given the differences described in lines 290 and following. We carried out experiments with the TFT without the

groundwater level as input (see our answer to remark 2). We will include these analyses in the supporting information and mention the results in the manuscript.

4.2

39) line 326-328 Here again, since the water level (WL) appears to be such an important feature, in comparision to static feature, i'm curious of what could be this correlation analysis if WL is removed of the input feature.

As the performance of the TFT model across all 5,288 monitoring wells was considerably worse without the groundwater levels as input feature, we decided not to pursue further analyses in this regard. It is important to note that our study includes a large number of monitoring wells with varying degrees of predictability, including many located in hydrogeological complex areas or influenced by anthropogenic effects.This contrasts with other studies that often focus on wells with inherently high predictive capacity (e.g. wells predominantly influenced by climatic factors), leading to higher NSE values. Nevertheless, there are still 884 wells in our study where the TFT model achieved an NSE >= 0.5 even without groundwater levels as an input feature.

Moreover, we believe that this topic is beyond the scope of this study. The main focus is on seasonal predictions, where the inclusion of historical measurements of the target feature as input is standard practice to enhance the models performance (see also our answer to remark 2). Removing this critical feature would not align with our study's objectives, which aim to evaluate the utility of advanced machine learning architectures in a realistic operational setting.

40) line 331 - what is the 'expected seasonality' ?

The term expected seasonality denotes the agreement of the hydrograph with lower groundwater levels in summer and higher groundwater levels in winter and is measured by the time series feature "seasonal behaviour". We describe this time series feature in section 2.4 (line 186 ff.).

41) line 331 "The highest identified [...]  a lower flashiness." Here again, it sound's like when WL evolution is 'simple' : sinusoidal variation with low flow/hight flow in summer/winter : the model performs. But I'll wait so much more from advanced DL models ! I suggest to mitigate or more discuss this case, with the perspective of the added-values of 'complicated' models compared to LSTM of even simpler models (exponential smoothing, VAR,...).

We will add and discuss examples where the models performed well for groundwater dynamics other than the expected seasonal patterns. See our answer to remark 38.1. Moreover, simpler baseline models often perform worse than more advanced methods and therefore provide limited additional insights. For instance, Gomez et al. (2024) used a sinusoidal curve with a precipitation trend as a baseline model, which demonstrated inferior performance.

42) line 335-337 One could also challenge the 52 weeks sequence here. The variability of this wells could be at lower frequency, i.e. needs a larger sequence

Generally, we agree with the reviewer. For a longer sequence length of multiple years longer validation and test periods would be needed to avoid overlapping periods. Unfortunately, many of the monitoring wells included in our study do not have sufficiently long time series data for this task. We will add a sentence to provide more context (line 336 ff.):

"Choosing a longer sequence length could improve predictions for those monitoring wells that deviate from the expected seasonality and exhibit lower-frequency variations (inert groundwater hydrographs). However, this was not feasible because of the lack of sufficiently

long time series for many monitoring wells to support longer sequence lengths, and accordingly longer validation and test periods."

43) line 340 : Porous aquifer here seems to denote also 'homogenous'. Would an highly compartimented aquifer made of porous sediment still refered as 'porous' ?

We followed definitions of the individual aquifer types in the hydrogeological map of Germany (HUEK250) and used them therefore in our explanations for the model performance as well. Principally, a highly compartmentalized aquifer made of porous sediment is indeed referred to as porous, but additional descriptors would be necessary to convey its degree of heterogeneity, which is not available for the HUEK250. Moreover, we do not claim that porous aquifers are homogeneous but in general rather less anisotropic than karst or fractured aquifers. We will clarify the sentence in line 340:

"This finding aligns with theoretical expectations that in porous aquifers and at moderate permeabilities groundwater flow is relatively slow and uniform and thus more predictable than in other hydrogeological systems such as karst and fractured aquifers, which exhibit highly anisotropic subsurface conditions with high flow velocities and heterogeneous groundwater dynamics (Bakalowicz, 2005; Hermans et al., 2023)."

44) line 361 Still the problem of WL as an input feature or not. This should be distinguished among cited references.

line 362 "However, these studies were conducted for a much smaller number of monitoring wells and the authors suggest that their models used the static features primarily as unique identifiers (Heudorfer et al., 2024; Li et al., 2022)." -> this is not the only difference. Here again I think that WL as an input feature plays a major role.

Had the performance of the previously mentioned TFT model without groundwater level as an input feature been better, a more direct comparison would have been possible. However, to the best of our knowledge, to date no other studies utilized both static and endogenous features for groundwater level predictions and evaluated the impact of the addition of static features. Therefore, we referred to the available studies in this context. We also note that we have addressed the difference to the cited studies afterwards in lines 366 and following. If the reviewer still wishes to address this difference to the cited studies we are open to modify line 360 and following:

"However, these studies were conducted for a much smaller number of monitoring wells with models trained without historical groundwater level measurements as input feature. The authors conclude that their models used the static features primarily as unique identifiers (Heudorfer et al., 2024; Li et al., 2022)."

45) Title 4.3 The Role of Static Features in Global Machine Learning Models. With the exeption of the last 4 sentences, all this part is dedicated to the usage of ML for groundwater level prediction. Title suggests that the general case will be discussed (which is not strictly the case).

We agree with the reviewer and will modify the title:

"The Role of Static Features in Global Machine Learning Models for Groundwater level Prediction"

46) line 366 "It is important to note that by using a validation set and various techniques such as dropout and early stopping to avoid overfitting the models were prevented from simply replicating historical groundwater levels." this is maybe part of an explanation. But i) it comes to late in the paper, and ii) ok for dropout, one can also think to pruning to achive such a goal, + but, I still believe that the fundamentaly autoregressive behavior of

groundwater makes the waterlevel itself as an input feature a big game changer, and comparision with models that do not take WL in inpute is then biaised.

Please refer to our answer to remark 44. Additionally, the measures we implemented to mitigate overfitting are described in Section 2.3

Conclusion :

47) Still the problem with groundwater level itself in the input feature that could lower the effect of dealing with static features...

data availability for static features, adequation between static features used here (mainly concerning soil/ surface cover), effect of the 1 km radius are missing here, despite beeing written and cleaverly discussed before. They should be added in the conclusion.

We will add these points raised by the reviewer also to the conclusion.